# Synchronous mid-Holocene marine and terrestrial deglaciation in the Ross Sea, Antarctica

Rebecca L. Parker [1,13] ✉, Christina R. Riesselman [1,2], Olivia J. Truax [1,3], Richard S. Jones[4], Jae Il Lee[5], Min Kyung Lee[5], Geraldine Jacobsen [6], Brad E. Rosenheim [7], Cristina Subt[7], Atun Zawadzki[6], Catherine Ginnane [8], Sebastian Naeher [9,10], Gavin Dunbar [11], Robert M. McKay [11], Richard Levy [8], Jocelyn Turnbull [8,12] & Kyu-Cheul Yoo[5]

The Ross Ice Shelf buttresses ice draining from both East and West Antarctica and its collapse could accelerate the loss of inland ice sheets, rapidly raising sea level. Documenting the location, timing and rate of past glacial retreat can help reveal processes driving rapid mass loss, informing projections of ice sheet responses to a warming climate. Here, we present a record of mid-Holocene ice retreat from the southwestern Ross Sea using facies succession and paired ramped pyrolysis oxidation $^{14}C/^{210}Pb$ chronology. This record shows rapid ice shelf retreat from 6.9-5.4 cal kyr BP, coeval with thinning of adjacent outlet glaciers. Our findings reconcile earlier discrepancies in terrestrial and marine reconstructions, and indicate that synchronous grounding line retreat from west of Ross Island to the Siple Coast at ~7-6.2 cal kyr BP was likely driven by warm-water incursions, a process active in parts of Antarctica today.

The Ross Ice Shelf buttresses ice draining from the East Antarctic Ice Sheet through the Transantarctic Mountains and from the West Antarctic Ice Sheet via ice streams; these two large catchments represent approximately 12 m of potential global sea level[1]. Understanding historical ice shelf stability is crucial for future sea level projections of ice sheet change, which are influenced by local bathymetry and ice-ocean interactions[2]. Although currently stable[3], model projections suggest that the Ross Ice Shelf will thin in the coming decades, causing grounding line retreat into the deep basins of West Antarctica and contributing meltwater to the global ocean[4–8]. However, the modern observational period is too short to confidently assess long-term responses to climate warming over centuries to millennia.

The nature of ice sheet palaeo-drainage and retreat in the Ross Embayment following the termination of the last glacial maximum (LGM) has significant implications for broader understanding of the dynamics of Antarctica's ice sheet margins and their sensitivity to climate forcing and local bed topography. Geological reconstructions provide insights into mechanisms of ice retreat and serve as benchmarks for ice sheet models (e.g.[5,9,10]). However, the timing and pattern of grounding line retreat remain debated due to the inconsistencies between marine and terrestrial records[10–15]. In the western Ross Sea, retreat was complex[9,13,14], with bathymetric highs stabilising the grounding line[16] and reconstructions of megascale glacial lineations recording shifts in ice flow direction[14]. Recent studies indicate that the

[1]Department of Geology, University of Otago, Dunedin, New Zealand. [2]Department of Marine Science, University of Otago, Dunedin, New Zealand. [3]School of Earth and Environment, University of Canterbury, Christchurch, New Zealand. [4]Securing Antarctica's Environmental Future, School of Earth, Atmosphere and Environment, Monash University, Melbourne, VIC, Australia. [5]Korean Polar Research Institute (KOPRI), Incheon, Republic of Korea. [6]Australian Nuclear Science and Technology Organisation (ANSTO), Lucas Heights, NSW, Australia. [7]College of Marine Science, University of South Florida, St. Petersburg, FL, USA. [8]Earth Sciences New Zealand, Lower Hutt, Wellington, New Zealand. [9]Department of Soil and Physical Sciences, Lincoln University, Christchurch, New Zealand. [10]School of Geography, Environment and Earth Sciences, Victoria University, Wellington, New Zealand. [11]Antarctic Research Centre, Victoria University, Wellington, New Zealand. [12]CIRES, University of Colorado at Boulder, Boulder, CO, USA. [13]Present address: Geography, Faculty of Environment, Science and Economy, University of Exeter, Exeter, UK. ✉e-mail: r.l.parker@exeter.ac.uk

grounding line retreated inland of its modern position along the Siple Coast during the mid-Holocene and subsequently readvanced[17–20], though this pattern has not been empirically constrained elsewhere in the Ross Sea. Model simulations reproducing this grounding line reversal depend on robust geological reconstructions to refine ocean heat forcings and solid Earth feedback parameterisations[17,18,21].

The largest discrepancies between marine and terrestrial reconstructions of Holocene ice sheet retreat based on geological deposits and between geological reconstructions and model simulations, occur in the southwestern Ross Sea. Reconstructions from marine sediment cores dated with foraminiferal radiocarbon ($^{14}$C) show that the grounding line retreated east of Ross Island by at least 8.6 cal kyr BP[10], reaching McMurdo Sound around ~7.5 ka[22]. Rapid ice surface lowering of Mackay and Mawson glaciers in the southwestern Ross Sea occurred between ~7.5 and 5 kyr BP, while raised beaches in McMurdo Sound (6.6 kyr BP) provide key indicators of the final timing of coastal ice retreat[23,24]. However, glacial lowering and modelled grounding line retreat lag empirical marine reconstructions in this region by ~2–5 ka[5,16,24,25]. These discrepancies may arise from age model uncertainty in marine records, complex bed topography, local subglacial conditions[5,16], or negative cold-cavity ice shelf feedbacks[21,26]. Resolving these intra-proxy and proxy-model differences requires further observational constraints on post-LGM grounding in this region.

Accurate $^{14}$C dating of marine sediments remains a major challenge for constraining the timing of ice retreat. Paired benthic and planktic foraminiferal assemblages from laminated strata are presumed to most accurately record the timing of sediment deposition, but are sparse or poorly preserved in Antarctica[10,27]. Consequently, bulk sediment acid-insoluble organic matter has commonly been dated to determine the age of sediment deposition. However, glacial processes are known to rework older carbon, either biasing bulk $^{14}$C towards older ages (e.g.[28]), or towards younger ages if sedimentary sequences are condensed and heavily bioturbated, as is the case in many Ross Sea cores[29]. Ramped pyrolysis oxidation (RPO)–$^{14}$C dating, which thermochemically separates different carbon pools in the sediment, addresses many of these limitations[19,20,27,30,31]. Through ramped pyrolysis, syn-depositional sedimentary marine organic matter from the water column is dated in isolation from pre-aged, transported material[32]. This approach is further strengthened by pairing with pyrolysis gas chromatography-mass spectrometry (Py-GC-MS) analysis, which identifies source organic compounds to enable preferential dating of splits containing primary photosynthetic products[31]. An additional challenge of dating marine sediments in the Antarctic is the large (~1100 years[23,33]) and variable[34] reservoir age, which must be accounted for when calibrating $^{14}$C ages to calendar years. However, application of additional radiometric techniques (U-Th disequilibrium; $^{210}$Pb) can provide independent age constraints on both depositional age and the magnitude of the radiocarbon reservoir correction.

Here, we present RPO $^{14}$C and $^{210}$Pb-constrained retreat histories that reveal synchronous mid-Holocene grounding line retreat and glacial lowering in the Ross Embayment, demonstrating the importance of bed topography and ocean forcing during retreat. Two sites, RS15-GC78 and GC80 are ideally located to assess the relationship between ice sheet grounding line retreat and glacier surface lowering because they are situated within the respective paleo-drainage paths of Mawson and Mackay glaciers, two glaciers with well-constrained ice lowering histories[24,35]. Two additional sites northeast of Ross Island, GC71 and GC72, provide a link to central Ross Embayment retreat[10] (Fig. 1). This network of sites allows us to document the timing of marine-based calving line retreat across this sector of the southwestern Ross Sea and compare it to adjacent terrestrial glacier thinning records with the support of regional ice-sheet modelling. Widespread mid-Holocene grounding line retreat is supported by the pattern of ice retreat in the Ross Sea Embayment; retreat in the region west of Ross Island was contemporaneous with grounding line retreat towards the Siple Coast.

## Results and discussion
### New estimates for ice sheet retreat
Ross Sea sediment cores reveal a stratigraphic succession from subglacial to open marine environments[10,22,36–38] (Figs. 1 and 2; Supplementary Figs. 1–4; Supplementary Table 1). Basal muddy or sandy diamicton facies (Dm and Ds, respectively) are overlain by bioturbated silty clay facies with common clasts (Mc) indicative of a proximal ice shelf grounding line environment NE of Ross Island at GC71,

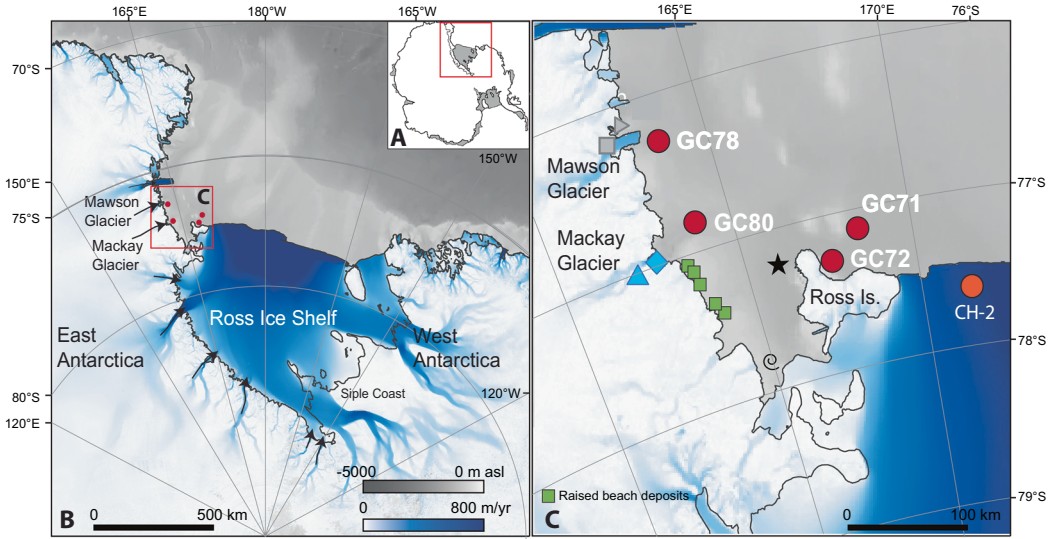

**Fig. 1 | Study context. A** Map of Antarctica; **B** the Ross Sea sector with modern ice flow velocity (colour scale[55]) and bathymetry (greyscale). Arrows denote ice flow direction from major glaciers along the Transantarctic Mountains and **C** Southwestern Ross Sea region highlighting the location of cores RS15-GC71, GC72, GC78 and GC80 marine sediment core sites (red circles), collected on the 2015 KOPRI Antarctic Cruise ANA05B onboard the R/V *Araon*. Other southwestern Ross Sea sites mentioned in text from the southwestern Ross Sea include Coulman High (CH)-2 (orange circle[10]), reworked shell (black star[22]); exhumed corals from the grounding zone (black spiral[39]), raised beach deposits (green squares[23]) and surface lowering sites from Mawson Glacier (grey square and triangle[35]) and Mackay Glacier (bright blue square and triangle[24]) glaciers. (Map generated using Map with Geo-MapApp (www.geomapapp.org/CC BY) using the Global Multi-Resolution Topography synthesis database[79]).

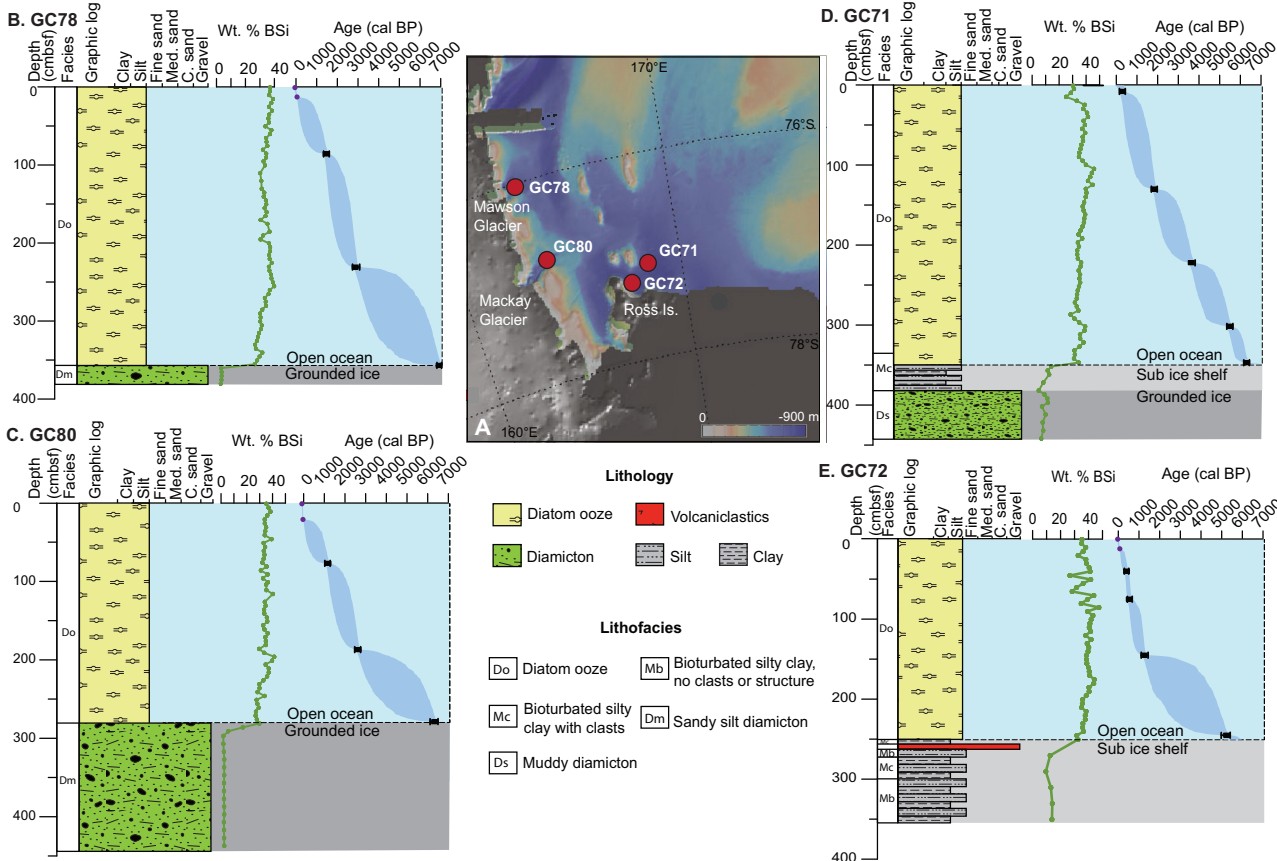

**Fig. 2 | Down-core evidence of ice retreat.** Down-core lithofacies, schematic graphic logs and wt% BSi records reflect retreat of the calving line across the core sites and the onset of open marine conditions in **A** the southwestern Ross Sea. For **B** GC78 and **C** GC80, ice retreat is recorded as the transition from diamicton to overlying diatom ooze with a pronounced increase in wt% BSi. **D** GC71 and **E** GC72 reflect the development of an ice shelf-like setting during retreat, followed by the onset of open marine conditions (diatom ooze). Grounding line retreat is not recorded in GC72 (diamicton absent). Calibrated [14]C dates shown by black squares and upper and lowermost [210]Pb dates shown by purple circles, both with 2 standard deviation error bars. Age models with 95% CI (shaded dark blue) were generated through the BChron package (version 4.7[77]) in R (version 4.3.3[80]). See 'Methods' section for radiocarbon age calibration details. (Map generated using Map with GeoMapApp (www.geomapapp.org/CC BY) using the Global Multi-Resolution Topography synthesis database[79] and ice thickness BEDMAP v2).

transitioning into a diatom ooze facies (Do) up to ~3.5 m thick, which represents open marine conditions. The transition from diamicton to Do is abrupt in GC78 and GC80, offshore of Mawson Glacier and Granite Harbour, respectively: both cores lack a sub-ice shelf facies. GC72 is located south of GC71 and displays a slightly different stratigraphy; no diamicton was recovered and a basal silty clay facies with intermittent clasts (Mc and Mo) transitions into Do. Dating the lowermost Do deposition, immediately above the transition between underlying diamicton or silty clay, reveals the onset of open marine conditions following calving line retreat at the core site.

We establish robust chronologies for cores GC71, 72, 78 and 80 by combining [210]Pb data with RPO-derived [14]C ages calibrated using the Marine20 curve and a regional marine reservoir correction (ΔR) of 609 ± 137 [14]C years[33,39]. Modern, or near modern core tops are confirmed by [210]Pb activity profiles for each core (Supplementary Text 1). RPO [14]C dates generally increase in age with pyrolysis temperature[32], but in our cores, the second temperature fraction ('split 2') systematically yields younger ages than the first ('split 1'), despite the higher temperature. This reversal, also reported elsewhere in the Ross Sea[30], likely reflects contamination of split 1 with volatile, older material that biases [14]C ages older (Supplementary Text 2; Source Data 1 and 2). Analysis of the organic compounds support this interpretation: pyrolysis gas chromatography mass spectrometry (Py-GC-MS) analysis on a subset of samples shows that split 2 typically displays a more diverse array of organic compounds (Supplementary Fig. 6), indicating that the lower-temperature fraction incorporates low-molecular

degradation products. Based on these results, we selected split 2 ages as the most accurate maximum age constraints available for cores GC71, 72, 78 and 80. For GC71 and 72, the uppermost [14]C dates calibrated with the regional ΔR alone are significantly older than their [210]Pb counterparts, suggesting a greater influence from residual detrital carbon contamination. To correct for this, we apply an additional local contamination offset (LCO) derived from the difference between the regional Ross Sea ΔR and a reverse-calibrated pre-bomb [210]Pb-[14]C pair in GC72 at 10 cm depth[40,41]. The resulting LCO of 870 ± 120 years was also applied to GC71 for which no equivalent pair exists (Supplementary Text 1 and 3). For GC78 and 80, [210]Pb chronology indicates that the uppermost [14]C dates are post-bomb and therefore cannot be reverse-calibrated to the Marine20 curve (Source data 1). We therefore use the regional ΔR for GC78 and 80 and exclude the uppermost [14]C dates in favour of the more precise [210]Pb chronology for the surface interval. Full methodological details and calibration rationale are provided in the Supplementary Information (Supplementary Text 1–3).

Our results show that open marine conditions developed in the southwestern Ross Sea following ice retreat between 6.9–5.4 (7.3–5.1, 95% confidence) cal kyr BP (Figs. 2 and 3; Supplementary Figs. 1–4, Source Data 2). At GC71, NE of Ross Island, a grounding-line-proximal to sub-ice-shelf setting developed as grounded ice retreated southward over the core site and open marine conditions were established by 6.4 (7.2–6.2, 95% confidence) cal kyr BP. Further south, at GC72, retreat was likely restricted by pinning around Ross Island, allowing an ice shelf setting to develop and persist before the establishment of open marine

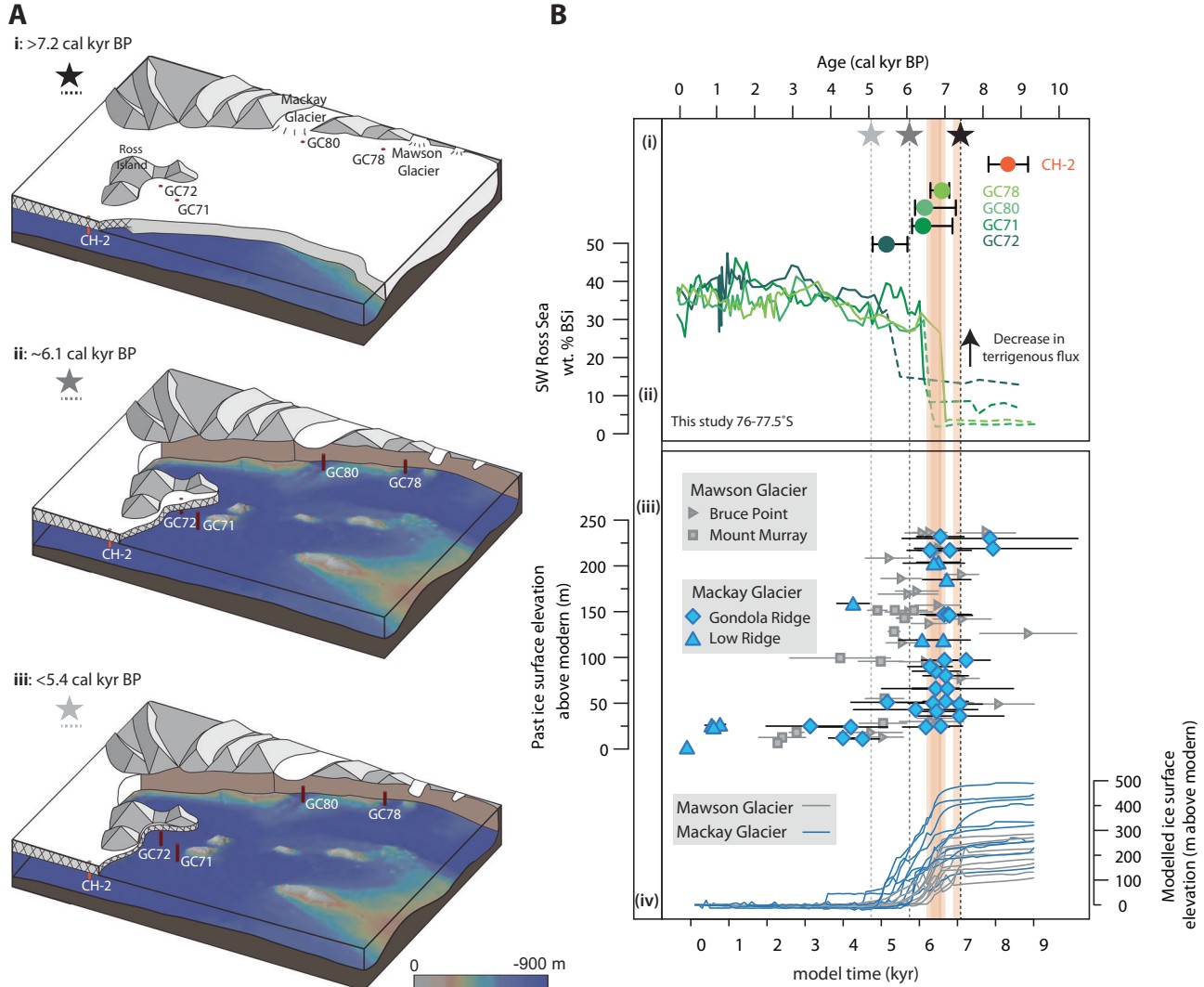

**Fig. 3 | Mid-Holocene marine and terrestrial ice retreat in the southwestern Ross Sea. A** Schematic illustration of post-LGM ice retreat timing and pace, reconstructed from lithological transitions in marine sediment cores (diamicton or silt/clay to diatom ooze) indicating ice retreat and the onset of open marine conditions. Hatched regions represent floating ice/ice shelf setting. (**i**) Preceding 7.2 cal kyr BP, grounded ice extended further north than Mawson Glacier and northeast of Ross Island. Core CH-2 records an ice shelf environment. (**ii**): By 6.1 cal kyr BP, the grounding line retreated into overdeepened troughs along the southwestern margin of the Ross Sea with open ocean conditions established at GC71, 78 and 80. An ice shelf environment persisted over GC72. (**iii**): By 5.4 cal kyr BP, all of our sites recorded open marine conditions. **B** (**i**) Retreat timing for GC78, 80, 71 and 72 (dark green to light green circles respectively), derived from RPO [14]C-[210]Pb ages used to constrain the onset of open marine conditions at each core site and grounded ice retreat at CH-2 (orange circle[10]). Horizontal lines indicate 95%

confidence intervals. (**ii**) Wt% BSi records for GC78, 80, 71 and 72 (dark green to light green as in (**i**)), a proxy for the onset of open ocean conditions at each site. (**iii**) Records of surface lowering (mean and 1 sigma uncertainty) at Mawson Glacier, with elevation transect at Mount Murray and Bruce Point[35] and for Mackay Glacier, with elevation transect at Gondola Ridge and Low Ridge[24]; (**iv**) Simulated ice surface lowering for Mawson (grey line) and Mackay (blue line) glaciers for a range of model runs[16]. The timing of schematic illustrations in **A** are denoted by stars and dotted vertical lines in (**B**). Model outputs are aligned with proxy data by matching simulated grounding line retreat at core sites (orange bars) to the recorded timing of retreat at the GC78 (Mawson) and GC80 (Mackay) core sites. Dashed lines in **B** (**ii**) denote sections of the BSi record that have no age control. (Bathymetry generated with GeoMapApp (www.geomapapp.org/CC BY) using the Global Multi-Resolution Topography synthesis database[79]).

conditions by 5.4 (6.0–5.1, 95% confidence) cal kyr BP. Along the western margin of the Ross Sea, cores GC78 and GC80 record a sharp transition from diamicton to overlying Do with no intermediate sub-ice shelf facies, indicating that open marine conditions were established quickly following local grounding line retreat. Ice had retreated beyond GC78 by 6.9 (7.1–6.7, 95% confidence) cal kyr BP and beyond GC80 by 6.5 (7.3–6.3, 95% confidence) cal kyr BP, into the overdeepened troughs of Mawson Glacier and Mackay Glacier, respectively.

### Inland ice drawdown coupled to rapid grounding line retreat
The new retreat dates provide a framework to better assess the characteristics and mechanisms of marine-based ice sheet retreat.

Combined with marine dates in the southwestern Ross Sea, recalibrated with the updated Marine20 curve[42] and the Ross Sea regional $\Delta R$[33,39], our results support a consistent spatial and temporal pattern of regional deglaciation. For example, the minimum age of ungrounding at Coulman High[10] becomes 8.6 (9.2–8.1, 95% confidence), while the modelled radiocarbon ages of raised beaches in the southwestern Ross Sea becomes ~8–7 cal kyr BP (Supplementary Table 3).

Our data show mid-Holocene grounding line retreat in the southwestern Ross Sea region is also consistent with geomorphological interpretations[13,14] and ice surface lowering estimates from cosmogenic studies[24,35]. Based on calibrated [14]C ages following the transition from (sub)glacial to open marine facies in cores GC71 and 72,

we constrain southwest retreat towards Ross Island to between 6.4 and 5.4 cal kyr BP. At the same time, the grounding line retreated 100–140 km towards the southwestern Ross Sea margin to reach the heads of the overdeepened troughs of Mawson and Mackay glaciers, with open marine conditions established by 6.9 and 6.5 cal kyr BP (cores GC78 and 80), respectively. At this time, rapid glacier thinning is recorded upstream of the core sites[24,35] (Fig. 3).

Our age constraints therefore allow us to assess the coupling between marine-based ice retreat and inland thinning, a relationship previously unresolved due to difficulties in aligning marine and terrestrial chronologies. This discrepancy is largest with the Coulman high core (8.6 cal kyr BP) and sites dated using the acid-insoluble organic fraction of bulk sediment. To assess the lead, lag or synchroneity of retreat and thinning requires (1) a causal relationship between recorded changes at the different sites and (2) reliable dates for the timing of these changes.

First, based on theoretical understanding of ice sheet retreat and on numerical modelling for the southwestern Ross Sea, we argue that there is a causal relationship between retreat at our core sites on the continental shelf and upstream thinning of the glaciers. Core sites GC78 and GC80 are located at the heads of retrograde slopes, where bathymetry within each trough deepens inland towards the present-day coastline. A grounding line positioned on such a bed slope is unstable and, once initiated, retreat will accelerate towards the coast until ice thickness at the grounding line reduces on an opposite upward-sloping bed and/or the lateral drag imposed by narrowing of the trough increases sufficiently to slow retreat–a process known as marine ice sheet instability[43]. Accelerated retreat into deeper bathymetry results in a substantial increase in ice discharge and associated ice surface drawdown of glaciers upstream (e.g.[24,44]).

Regional-scale modelling supports this coupling between marine-based retreat and upstream ice drawdown upstream[16] (Supplementary Fig. 7). Glacier retreat is initially slow as the grounding line is pinned to seamounts to the north of Ross Island. While ice surface lowering at Mawson and Mackay glaciers was first initiated by ice ungrounding from this shallow bathymetry, the fastest rates of glacier thinning occur in the model runs once the grounding line passes our core sites, over the retrograde bed slope (Fig. 3B). This thinning happens first at the downstream sites of Mawson and Mackay glaciers, 30–45 km from the core sites and later at the upstream sites, ~40–60 km from the core sites. We therefore suggest that retreat from the core sites led to rapid thinning at the glacier sites.

Second, we argue that marine ages for the timing of retreat and terrestrial ages for the timing of thinning are reliable. Traditionally, there has been concern about directly comparing dates from different types of evidence due to some systematic errors that can be hard to quantify, which would impact the absolute age of retreat or thinning. As a result, studies in the southwestern Ross Sea have typically referred to approximate timings from neighbouring marine or terrestrial records without direct comparison of the absolute timings (e.g.[10,35]).

For marine records, where the timing of grounding line retreat is based on $^{14}C$ dating of sediments, the main issues have been a potentially inaccurate marine reservoir correction and contamination from ancient detrital carbon in bulk $^{14}C$ dates. Typically, the marine reservoir correction is approximated from a regional or continent-wide average that may not be representative of the site[39,45], or the combined influence of both the marine reservoir and detrital carbon contamination is estimated from dated sediment at the site (e.g.[46]). However, as discussed above, by combining $^{210}Pb$ data with RPO-derived $^{14}C$ ages, we reduce contamination by ancient detrital carbon and calibrate using the Marine20 curve and a Ross Sea average ΔR value of $609 \pm 137$ $^{14}C$ years[33,39] to produce more accurate timings for the establishment of open marine conditions at the core sites.

For terrestrial records, where the timing of glacier thinning is largely based on cosmogenic exposure dating, the main systematic error has been the cosmogenic production rate. This is mostly because Antarctica has specific atmospheric pressure effects and there are no local calibrating production sites[47]. But, due to relatively recent developments in production rate estimation and scaling[48,49], exposure ages are now much more accurately calculated, including improved quantification of the production rate uncertainty. However, other factors could impact the absolute timing of thinning from cosmogenic exposure dating. Accounting for any glacial isostatic adjustment, subaerial erosion of the surface or snow cover would make the exposure ages older. The original glacier thinning studies assessed the influence of glacial isostatic adjustment to be <100 years, surface erosion to be negligible due to very low erosion rates in Antarctica and snow cover to likely be minimal due to high melt from the low albedo of rock and the selection of samples from wind-swept areas[24,35]. Conversely, accounting for cosmogenic inheritance would make the exposure ages younger. Inheritance is an issue for many sites across Antarctica due to insufficient subglacial erosion during periods of ice cover, often leading to age overestimation by several 10–100s kyr[50]. Minor inheritance in some samples may explain scatter in the exposure age transects, but we consider the overall effect to be negligible for each site as the samples are Holocene-aged. We therefore consider the terrestrial records at Mawson and Mackay glaciers to provide reliable estimates for the timing of glacier thinning.

We quantify the coupling between grounding line retreat and inland thinning by focusing on Mackay Glacier, which has the most precise constraints on the period of rapid thinning, with the timing of its onset and cessation statistically estimated[24,51]. Rapid thinning occurred between 7.1 ka (5.8–8.1 ka, 95% confidence) and 6.1 ka (3.6–6.9 ka, 95% confidence), whereas downstream retreat at GC80 occurred at 6.5 (7.3–6.3, 95% confidence) cal kyr BP. Although synchrony cannot be ruled out at 95% confidence, a 600 year (−1.5–1.8 cal kyr BP at 95% confidence) offset is supported by modelling. The results show rapid thinning beginning a few hundred years before retreat at the core sites (Fig. 3B), with further thinning accelerating after the grounding line migrated inland from site GC80 (Fig. 1). Our data are consistent with rapid thinning ending 1.5 kyrs (or up to 3.7 kyr at 95% confidence) after retreat from the offshore core sites (Fig. 3B). Given the distance between marine and terrestrial sites, these data provide evidence that rapid centennial-scale thinning can be triggered from grounding line retreat that occurs over 60 km downstream, with the rate of ice drawdown likely peaking due to bathymetric controls relating to marine ice sheet instability processes (e.g.[52]).

## Topography controls ice sheet retreat

Our records provide new constraints on deglaciation patterns in the southwest and broader Ross Sea. Following an early linear model of deglaciation[11] (Fig. 4A), recent models infer that seabed topography substantially influences ice retreat patterns, resulting in distinct regional variations e.g.[9,10,12–14,53–57]; Fig. 4B).

Here, we illustrate a retreat scenario (Fig. 4C) with more distinct spatial and temporal regional variability than has previously been postulated from empirical data (Fig. 4A, B) and captured in model simulations (e.g.[5,25]). During the LGM, grounded ice filled the Ross Sea to the continental shelf edge, subsequently retreating to reach its current position sometime between 6.3 and -1.7 ka BP[12,18,20,58]. Our results suggest that the embayment in grounded ice, bounded by Ross Island to the west and the Siple coast, formed rapidly, at the latest, between ~8.1 (site CH-2[10]) and 6.2 (Mercer Subglacial Lake[20]) ka BP. This rapid retreat occurred as the RIS cavity was reaching its maximum size ca. 6–7 ka BP[20], which coincided with enhanced Ross Sea-sourced glacial meltwater along the Adélie Land coast between 6.5 and 4.5 ka BP[26], before a slowdown of retreat and regrowth of the ice sheet during the late Holocene[20,59].

In the southwestern Ross Sea, the timing of retreat recorded in all our cores supports the critical role of seabed topography in both enhancing and limiting the rate of grounding line retreat. The earliest

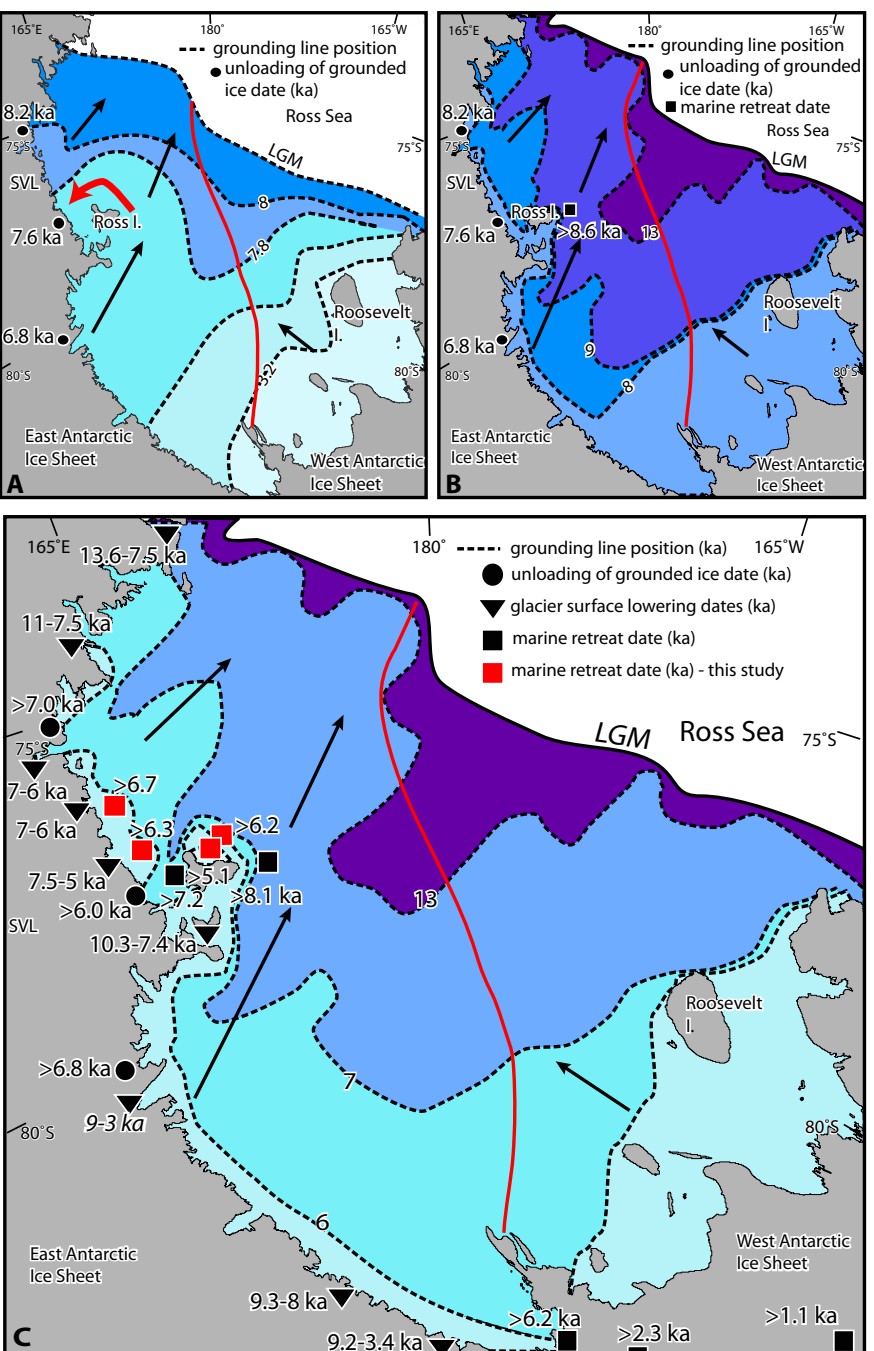

**Fig. 4 | Progression of deglaciation theories in the Ross Sea, Antarctica. A** Early model of deglaciation. Timing of deglaciation is constrained by terrestrial geological data (black circles[61,81]) in Southern Victoria Land, invoking a linear southward migration of the grounding line along the western Ross Sea margin. **B** Earlier retreat in the central Ross Sea, with retreat along the western margin occurring later. Timing is based on combined geological and modelling constraints[9,10]. **C** Updated pattern of retreat based on recalibrated existing dates from (**B**), new constraints from the Siple Coast[18,20] (black squares), cosmogenic-based timing of accelerated thinning along the Transantarctic Mountains with dates in italics where thinning was relatively gradual or rate is unclear[15,24,61,62,82,83] (Supplementary Table 4; inverted triangles) and from this study in the southwestern Ross Sea denoted by red squares.

Note the marine retreat ages are the minimum 95% confidence interval value and therefore the youngest statistically likely timing of retreat at these sites (Supplementary Table 3). Grounding line retreat was contemporaneous in the southwestern Ross Sea west of Ross Island and at the Siple Coast alongside broadly concurrent surface lowering of glaciers in the same region. Darker blue contour fill indicates earlier grounding line retreat. Red line is the tectonic boundary between the western and eastern Ross Sea. Arrows denote paleo-ice flow direction. **A**, **B** adapted from Lee et al.[13] with permission from co-author R. McKay, in accordance with GSA copyright policy (https://www.geosociety.org/GSA/gsa/pubs/guide/copyright.aspx).

age of retreat is recorded in deep bathymetry to the east of Ross Island (CH−2) and retreat in glacier troughs (GC78 and 80) is shown to be synchronous with rapid inland thinning. This supports the idea that marine-based ice sheet retreat is fastest in areas of deeper water, with the potential to be unstable and irreversible. Conversely, retreat should be slowest in areas of shallow water, particularly where multiple pinning points exist, which explains the later retreat just north of Ross Island (GC71, GC72) relative to that east of Ross Island (CH-2).

Despite these local controls on grounding line retreat, mid-Holocene retreat between 6.9 and 5.4 ka BP was contemporaneous at

our sites west of Ross Island and along the Siple Coast[15,19,20,60–62]. To the south of Ross Island, accelerated thinning of outlet glaciers started prior to retreat at our sites (>-9 ka BP) with the exact timing varying between glaciers along the Transantarctic Mountains (Supplementary Table 4). Glacier thinning was likely initiated by the southward migration of the grounding line in the central Ross Embayment, with the timing of accelerated thinning concurrent with grounding line retreat east of Ross Island (Coulman High[10]). The timing implies rapid retreat of ~800 km across the embayment towards the Siple Coast[18,20], likely due to a sustained forcing, with limited stabilising influences from shallow topography or an ice shelf. The retreat resulted in dynamic thinning of glaciers many 10s and possibly 100s of kilometres inland. Subsequently, retreat and thinning occurs in the southwestern Ross Sea west of Ross Island, further supporting our assessment that local seabed topography played a fundamental role in controlling the timing and magnitude of catchment-scale deglaciation (Fig. 4C).

Mass loss is currently observed in many marine-based Antarctic ice streams, with fastest rates of dynamic thinning (>3 m yr$^{-1}$) observed within ~50 km of the grounding line[63]. Here we show that such dynamic thinning can persist over multiple centuries, with widespread ice drawdown propagating inland. However, in areas of shallow water and complex seabed topography, our work highlights that deglaciation is substantially slower, potentially delaying ice sheet retreat by millennia. While bed topography will dictate the rate and timing of ice drawdown, the synchronous nature of deglaciation of the Ross Sea Embayment during the mid-Holocene indicates a sector-scale driver of retreat (e.g.[21]). This deglaciation was likely triggered by sub-shelf melt from incursions of warm Modified Circumpolar Deep Water[21,64] and our findings demonstrate the potential synchronous sector-scale ice loss that could occur in the areas of Antarctica experiencing these warm water incursions today.

## Methods

### Sample collection
Gravity cores RS15-GC71, 72, 78 and 80 were collected from the southwestern Ross Sea by the R/V *Araon* during the ANA05B Cruise in 2014/2015. The cores were split, imaged and described at the Korea Polar Research Institute (KOPRI) in August 2015.

### Geochemistry
Biogenic silica (wt% BSi) was measured at 5 cm intervals on all cores at the University of Otago using an alkaline extraction spectrophotometric method modified from refs. 65,66 using a Shimadzu UV-1800 spectrophotometer. Samples were frozen and freeze-dried before being homogenised using an agate mortar and pestle. Replicates were measured after every 10th sample and each analytical run contained one blank and one internal standard derived from the sample set. The average standard deviation of the nine internal replicates is 0.55%.

### Grain size
Grain size data were analysed at 5 cm intervals at KOPRI on samples treated with $H_2O_2$ and HCl to remove organic matter and $CaCO_3$, respectively. The coarse fraction (>63 µm) was analysed with sieves to determine the distribution of gravel to very fine sand sizes. The fine fractions were analysed for grain size distribution using a Micrometric Sedigraph III 5120. Grain size statistics follow the methods of ref. 67 and the lithological classification scheme follows ref. 10.

### Core scanning
Unsplit cores were logged after collection onboard the R/V *Araon* using the Alfred Wegener Institute Geotek Ltd Multi-Sensor core logger. This device measures wet bulk density by gamma ray attenuation, compressional (p-) wave velocity and magnetic susceptibility (e.g.[68]). Values of Cs-137 gamma ray attenuation were empirically calibrated to wet bulk density using variable Al-water mixtures to define a density

gradient, with pure water having a density of 1.0 g cm$^{-3}$ and Al 2.7 g cm$^{-3}$. P-wave velocity was calculated from p-wave travel time through a water standard of known temperature, which allowed for calculation of the p-wave travel time offset (PTO) representing the transit of the acoustic wave through core liner and electronic delay. The PTO is then subtracted from subsequent measurements of p-wave travel time through sediment with core liner to calculate the transit time through sediment of known thickness. Magnetic susceptibility was measured with a loop sensor using a Bartington MS-2 metre set in 'SI' measurement mode.

During the processing of the data, it became clear that the core thickness sensor was experiencing drift over time. Because both density and velocity data are dependent on accurately knowing the core (and thus sediment thickness once liner thickness is subtracted), we institute a correction on the basis that the drift was at a constant rate and as measurements are made systematically down core applying a linear thickness correction over the length of the core was appropriate. We ran the same water standard before and after each core was logged and the deviation from known thickness values for this standard were used to determine the magnitude of correction.

### Age model
**Lead-210 dating.** $^{210}Pb$ dating of sediment was conducted in the Environment Radioactivity Measurement Centre at ANSTO, Sydney. The alpha-particle spectrometry technique was used and is preferable over gamma-ray spectrometry as it requires smaller samples and gives lower detection limits. The samples were processed to prepare Po and Ra alpha sources. The activities of $^{210}Po$, the proxy for total $^{210}Pb$ and $^{226}Ra$, the proxy for supported $^{210}Pb$, from these sources were determined by alpha-particle spectrometry to calculate unsupported $^{210}Pb$ activities.

Approximately 2 g of homogenised sediment were analysed for each sample from 4 to 8 depths in the upper 20–40 cm of sediment for each core (see Supplementary Text 1, Supplementary Fig. 5 and Source Data 1). The chemical separation of Po is achieved through acid digestion to extract metals in the sediment into solution, complexation, pH adjustment of the solution and auto-deposition onto a silver disc. This disc is analysed by alpha-particle spectrometry. Following the removal of Po from solution, Ra is concentrated by co-precipitation, collected by colloidal precipitation of barium sulphate on a membrane filter and analysed by alpha-particle spectrometry. Unsupported $^{210}Pb$ activities, calculated from the difference between $^{210}Po$ and $^{226}Ra$ activities, in the sediment were used to determine sediment ages (see Source Data 1 for $^{210}Pb$ data) using the constant flux constant sedimentation $^{210}Pb$ dating model[63,64].

### Ramped pyrolysis oxidation (RPO) radiocarbon dating
Sediment samples for RPO radiocarbon dating were taken from the Do section of each of the four cores (16 samples in total), selecting for 1 cm horizons containing well-defined laminae to reduce the effects of bioturbation (Source Data 2). Samples were dried, homogenised and acid-treated with dilute HCl (1N) to remove carbonate minerals for at least 1 h until effervescence ceased. Acid was removed by centrifuging and rinsing with deionised water until the supernatant reached pH of 7[32].

These samples were prepared using RPO at the College of Marine Science, University of South Florida (USF). For each sample analysed at USF, we followed established protocol from ref. 32. The sample size was determined by using the %TOC required to yield ~100 µmol $CO_2$. Using a standard pyrolysis temperature ramp of 5 °C min$^{-1}$, complex mixtures of organic carbon in sediment samples are separated by leveraging the thermochemical stability of the acid insoluble organic matter. These pyrolysates are then combusted into $CO_2$[27,32,45].

Sample $CO_2$ gas aliquots were collected sequentially at intervals of approximately 10, 12, 25, 25, 25 µmol and sealed in pre-combusted and evacuated borosilicate ampoules with 150 mg of copper oxide and 1 cm of silver wire. The lower temperature aliquots are small to minimise the amount of detrital $CO_2$ incorporated. Ampoules with the

sample gas, copper oxide and silver wire were heated in a furnace at 525 °C for 2.5 h to remove sulphur, if present, prior to graphitisation.

Radiocarbon measurements of the $CO_2$ gas samples were performed at the Centre for Accelerator Science, ANSTO, Sydney[69]. The first two ampoules (containing 10 and 12 μmol of $CO_2$) for each sample and a series of blanks comprising three each of radiocarbon-dead material (graphite) and modern standard material (SRM 4990C oxalic acid, OxI) were analysed at the ANSTO for radiocarbon dating using the ANTARES accelerator[70]. This accelerator was used for its ability to process low carbon mass samples with high precision. The $CO_2$ from these samples, blanks and standards was converted to graphite by reduction with excess hydrogen over iron catalyst to produce targets for AMS measurements[71]. The graphite was pressed into cathodes and measured using the ANTARES accelerator[72], followed by $\delta^{13}C$ measurements on the graphite using IRMS to account for natural isotopic fractionation. Initial measurements were normalised to a $\delta^{13}C$ of −25‰ relative to the PDB standard.

RPO-AMS measurements were also performed on three additional samples at the Rafter Radiocarbon Lab (RRL) AMS facility at GNS Science, New Zealand following procedures established by ref. 31 in collaboration with USF. One distinction, however, is that RPO samples are split according to inflection points in the RPO thermographs and quantified after separation at RRL. The $CO_2$ is then recombusted (500 °C, 4 h with silver wire and copper oxide), graphitised and measured for radiocarbon content[73,74].

### Ramped pyrolysis-gas chromatography-mass spectrometry (Py-GC-MS)

Ramped Py-GC-MS was used to determine the composition and quantify the sources of organic carbon in the RPO splits of the three samples analysed at RRL (e.g.[31] Supplementary Text 2; and Supplementary Fig. 6). Samples were from a single horizon from cores GC72, GC80 and GC71 (40, 76 and 301 cm, respectively) and analysed at the GNS/VUW Organic Geochemistry Laboratory at GNS Science, New Zealand using a microfurnace-based Frontier Laboratories double-shot pyrolyser coupled to an Agilent GC-MS system, as reported in ref. 31. In brief, for full, rapid-ramp pyrolysis, samples were ramped from 100 to 650 °C at 50 °C min$^{-1}$. Then, incremental, partitioned ramped pyrolysis was used to replicate the ramped pyrolysis split temperature ranges for those samples. The resulting pyrolysis products were analysed by GC-MS analysis and relative proportions of compound classes determined (pyrroles, furans, alkanes and alkenes, toluene and other aromatics, thiophenes, polycyclic aromatic hydrocarbons and alkylbenzenes).

### RPO data processing

All $^{14}C$ data collected by RPO must be blank-corrected for contamination introduced into the sample during processing[75]. The correction process is consistent for samples prepared at USF and RRL, but the magnitude of the correction differs due to small differences in the reagents and systems used in the two laboratories. At the time of analysis of these samples, 109 geological graphite (devoid of radiocarbon) and Oxalic Acid I (OxI), nearly modern levels of radiocarbon[76] had been processed to assess the USF system's blank contamination. We calculate the mass of blank contamination which contains no $^{14}C$ (dead blank mass; 1.61 ± 0.9 μg) and the mass of the blank contamination which contains modern levels of $^{14}C$ (modern blank mass; 2.46 ± 1.6 μg) using a subset of this data set (19 datapoints) most appropriate for samples prepared in 2015 (see discussion in ref. 30). An additional (but smaller) blank correction was applied based on the contamination introduced during graphitisation and AMS measurement at ANSTO and the magnitude of these corrections was derived from materials introduced directly into the graphitisation system ($^{14}C$-free Kapuni $CO_2$ gas and OxI prepared in a large $CO_2$ batch).

For RPO performed at RRL, the contribution of modern carbon contamination is monitored with a $^{14}C$-free kauri wood and the contribution of dead carbon blank is measured with the OxI modern standard. Blanks encompass the complete process, including contamination from pre-treatment, RPO preparation, graphitisation and AMS measurement. Unlike USF, RRL observes a time-dependency in the blank contamination as part of the RPO process, which is also accounted for in the blank correction[31].

### Age model development

The age models for cores GC71, 72, 78 and 80 are based on a combination of RPO-$^{14}C$ radiocarbon and, where available, core-top $^{210}Pb$ dates. Calibration of $^{14}C$ dates to calendar years before present (cal yr BP) was performed using the BChron R package[77] and the Marine20 calibration curve, which includes a time-varying global reservoir age of ~603 years[42]. The current Ross Sea-wide ΔR of 609 ± 137 $^{14}C$ years is used for all dates[33,39]. We further apply a LCO for GC71 and GC72. Further detail on the radiocarbon reservoir age methods is in Supplementary Text 3.

### Glacier thinning from cosmogenic exposure dating

The timing of glacier thinning at Mackay and Mawson glaciers is determined from cosmogenic exposure dating of glacial cobbles at sites adjacent to the glacier[24,35]. The exposure ages were calculated for all sites using iceTEA[78], with the global $^{10}Be$ production rate calibration dataset[49] and nuclide-specific scaling model 'LSDn'[48]. While using a Southern Hemisphere production calibration site would make the exposure ages ~1 kyr older[24], we consider the global dataset and nuclide-specific scaling to produce the most appropriate exposure age estimates based on available evidence in Antarctica. The rate, onset and cessation of the rapid thinning recorded at Mackay Glacier were taken from published estimates[51], which also used the global production calibration dataset and 'LSDn' scaling.

### Modelling

We use regional-scale ice sheet modelling to assess the coupling relationship between the recorded grounding-line retreat and inland glacier thinning[16]. Modelling was carried out for the southwestern Ross Sea using the finite element model Úa, which solves for ice flow in the grounded and floating portions of the ice sheet simultaneously. In ref. 16, experiments were designed to capture grounding-line migration and corresponding upstream ice thickness change for 9–0 kyr ago, using two different climate forcings and five different combinations of the basal slipperiness and ice rate factor. To determine the likely relative timings between retreat and thinning in the southwestern Ross Sea, we use the five simulations from the experiment that had enhanced early-to-mid Holocene ocean thermal forcing (E2 in ref. 16), as these simulations best fitted the timing and rate of recorded thinning at Mackay and Mawson Glaciers.

## Data availability
All data generated or analysed during this study are included in this published article (and its Supplementary information files). Source data are provided with this paper.

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

## Acknowledgements

We acknowledge the exceptional science party, technicians and crew of expedition ANA05B of RVIB *Araon* for facilitating the sediment coring that provided the foundation for this work. This work was supported by Korea Polar Research Institute (KOPRI) grant funded by the Ministry of Oceans and Fisheries (KOPRI project No. PE25090), the New Zealand Ministry of Business, Innovation and Employment (MBIE) Past Antarctic Climate and Future Implications Programme (contract C05X1001), the MBIE Antarctic Science Platform (contract ANTA1801) and the MBIE Global Change Through Time research program (contract C05X1702). R.L.P. gratefully acknowledges additional funding from a University of Otago Masters Scholarship, an Antarctica New Zealand Post Graduate Research Scholarship and an Australian Institute for Nuclear Science and Engineering (AINSE) Ltd Postgraduate Research Award (ALN-STU12075) for radiocarbon and lead-210 dating. Additional support to C.R.R. was provided by a University of Otago Research Grant and a L'Oréal-UNESCO For Women in Science Australia and New Zealand Fellowship. O.J.T. was supported by a University of Otago PhD

Scholarship and an Antarctica New Zealand Sir Robin Irvine Doctoral Scholarship. R.S.J. was supported by the Australian Research Council under grant numbers DE210101923 and SR200100005 (Special Research Initiative, Securing Antarctica's Environmental Future). R.M.M. was funded by the Royal Society of New Zealand Marsden Fund contract MFP-VUW2207. Finally, the authors acknowledge the teams at the Rosenheim Lab, USF, ANSTO and the Rafter Radiocarbon Laboratory, Earth Sciences New Zealand, New Zealand for radiocarbon sample processing and measurement.

## Author contributions

K.-C.Y., R.L, C.R.R., R.M.M. and R.L.P. secured funding for various aspects of the research. K.-C.Y. led the R/V *Araon* expedition and J.I.L., M.K.L., G.D., C.R.R., R.L.P. and C.S. collected, scanned and described the sediment cores, with additional core descriptions by R.M.M. and R.S.J. C.R.R. designed this study. R.L.P. carried out the $^{210}$Pb dating, RPO $^{14}$C dating, age model development, geochemical analyses and data analysis. A.Z. assisted with $^{210}$Pb dating, B.E.R. and C.S. assisted with RPO $^{14}$C dating at USF and G.J. assisted with AMS measurements at ANSTO. C.G. and J.T. performed additional RPO $^{14}$C dating at RRL. S.N. contributed sample preparation, Py-GC-MS analyses and data interpretation. O.J.T. contributed to Py-GC-MS RPO interpretation and age model development. M.K.L. and J.I.L. performed grain size analyses. R.S.J. produced the glacier modelling and comparison with marine records. R.L.P., C.R.R., O.J.T. and R.S.J. drafted the manuscript. R.L.P., C.R.R., O.J.T., R.S.J., J.I.L., M.K.L., G.J., B.E.R., C.S., A.Z., C.G., S.N., G.D., R.M.M., R.L., J.T. and K.-C.Y. discussed the results and contributed to the final version of the paper.

## Competing interests

The authors declare no competing interests.
