## [Transparent Peer Review File · Nature Communications]

Synchronous mid-Holocene marine and terrestrial deglaciation in the Ross Sea, Antarctica

Corresponding Author: Dr Rebecca Parker

Version 0:

Reviewer comments:

Reviewer #1

(Remarks to the Author)

The authors present here new marine radiocarbon data based on ramped pyrolysis (RP) from the western Ross Sea that bear on the timing of ice recession following the last glacial maximum. They integrate these data with others from the region, as well as cosmogenic nuclide ages of ice thinning of nearby outlet glaciers, to present a synthesis of the timing and pattern of Holocene ice retreat.

The authors are to be commended in their attempt to bring together a wide range of different data types in their synthesis. However, this paper suffers from issues that should be addressed prior to publication.

In my opinion, the main issues can be summarized as:

- 1) The new delta R is not justifiable both in its calculation and application. It also leads to inconsistencies in the dataset that would not exist if they used a standard delta R for this region (which is well-known).
- 2) The paper needs more specific information on how calculations were done and on how ages presented in the text were determined. There are also numerous inconsistencies and/or typos in the text and figures that need resolving prior to publication.

1. Delta R

Marine dates in this study are based on ramped pyrolysis (RP), which aims to separate out different carbon fractions that have different origins, with the thought being that the more labile fractions reflect more recent marine carbon deposition and the more refractory components issuing from reworked ancient material. Ideally, one would be able to isolate only the recent fraction in order to obtain accurate ages for Ross Sea sediments. Dating has come a long way since the early studies in this region using simple TOC – which produced grossly old (and erroneous) ages for deglaciation. However, RP is not perfect and 100%, confident separation of the fractions cannot at present be achieved. This has been noted in prior studies (e.g., Venturelli et al., 2023) and appears to be the case in this present study. In the Supplemental Information (SI) the authors do a good job of explaining which fractions are thought to be autochthonous and which are reworked. There is also a substantial number of compounds that could be from both sources. Table S2 is especially helpful. However, from a combination of Table S2 and Fig. S6, one can see that even in the preferred Split 2, only roughly half of the compounds can be confidently attributed to autochthonous sources. Another ~40% could be either and about 7-10% are from refractory sources (ancient). Thus, the resulting radiocarbon dates from this split have a likely component of old material of at least 7-10% and maybe as much as 50%. This must affect the age, which means that these RP ages should be considered maximum ages only. Lines 225-226 of the SI refer to the resulting radiocarbon ages correctly as “maximum ages,” a point that seems lost when calculating and applying the delta R in the main paper.

If these RP ages are only maximum limiting ages, then they cannot be used in the manner proposed here to derive a marine delta R, because they are not the true radiocarbon age of the core top. At best, it is a maximum possible delta R for that place and time. As the proposed delta R (1480 yrs) calculated by this dubious manner is more than twice that of other measured values in the area (~610 yr; recalculated in Gao et al., 2022 and Hall et al., 2023) – in an area that already has one of the largest delta Rs in the world – it seems likely that the delta R proposed here is too large and should not be used.

The authors suggest that the commonly used delta R value of ~610 yr for McMurdo Sound (site of the original Hall et al. 2010 U/Th measurements from corals) may be younger than that for their site east of Ross Island due to ocean circulation differences. However, similar delta R values also exist for Terra Nova Bay to the north (Hall et al., 2010), near the Siple

Coast grounding line (Venturelli et al., 2023) nearly 1000 km away, and in the entire circum-Antarctic region based on marine shells collected prior to 1940 (when recalculated; Berkman and Forman, 1996), suggesting that this ~610 yr delta R value has robust regional significance. A quick look at the Marine Chrono database reveals that the average delta R for all Ross Sea samples is ~620 yrs. Thus, this delta R of 1480 yr calculated here is more than 3 sd from the mean and needs to be reevaluated critically. The authors do at one point mention that their delta R might be only applicable to their local site, but they then nevertheless apply it widely in the region, including to their two core sites on the western coast, one of which (McKay) is at the north end of McMurdo Sound. Applying their large delta R to these two core sites (GC78 and 80) results in the core top ages dating well into the future. All of this suggests that the proposed delta R is simply too large, most likely because no matter how good the RP and how much of an improvement over prior methods (which it is), there is some component of old material still left, which is affecting the age, making the delta R calculation an overestimate.

The authors suggest that the exact ages (which differ by nearly 1000 yrs) don't matter, just the pattern. I disagree. The pattern has been emerging for some time, with early models of a "swinging gate" giving way over the last decade to "saloon doors" with the center of the embayment opening up first. The approximate timing of retreat – mid-Holocene event - also has been known for some time. It is at the point where we need precise chronology and reliable onshore-offshore correlations – which is why exact ages do matter.

2. Need for specific information and correction of inconsistencies.

This paper needs more detailed methodological information. Readers are left at critical points wondering exactly how certain ages were calculated, whether the number referred to is actually a delta R (as opposed to a reservoir effect), what production rates and scaling were used for cosmogenic ages, were internal or external errors used in comparisons, etc... There are also several instances of figures not matching captions and text not matching tables. Line examples for all of these are below.

Line comments:

35-36 – I am not sure that the results as interpreted here reconcile long-standing differences. In fact, by introducing such a large reservoir effect, the marine ages are now younger, in my opinion, than the cosmogenic ages at McKay Glacier or the coastal marine deposits of McMurdo Sound.

69 – Is Ross Island really the southwestern Ross Sea? There is a lot more Ross Sea south of Ross Island.

98 – this reference is out of date, with more recent papers and the Marine Chrono database suggesting 1100 yrs (e.g., Hall et al., 2010).

Fig. 1 – needs lat./long.

Fig. 2 – larger text would make this more readable

159 – I assume from the extended data table that the authors compared the samples at 10 cm depth and not at the core top. There needs to be specific information about which two dates were compared, if not here then definitely in the SI. This gets back to Point 2 above.

160 – how did you reverse calibrate? Neither of the cited references give you how. Perhaps cite Reimer and Reimer 2017 if you followed their method.

161-163 As mentioned, this delta R should be treated with a large dose of skepticism. It is not "slightly older" than that reported from the corals of 609 yr. It's 900 yrs larger, which in the world of deltas R is a massive difference.

165 – how can you be sure of minimal contamination when the fraction that is confidently autochthonous is relatively small and there is a non-zero refractory component, which is almost certainly old?

168 – this argument would be more convincing if similar reservoir effects to McMurdo Sound were not seen elsewhere (Point 1 above) = This implies to me that the reservoir age from site GC72 is either too old because of contamination by old material or that it is very site and time specific and should not be applied elsewhere.

171 – Because I am not confident in the applied reservoir correction, I am not confident in the ages in this paragraph for ice retreat.

I think it is also important to stress that your dates are for the onset of open marine conditions, which is not necessarily that of grounding-line retreat. It's a minimum age for grounding-line retreat. This point gets lost at various stages of the paper. Perhaps the use of "before" rather than "by" (e.g., line 181, 182) would help make this point. Hall et al. (2010) showed that coral was already growing in southern McMurdo Sound by ~ 6 ka from U-series dating. Thus, McMurdo Sound had opened up before that time, which is older than the dates you propose for sites 78 and 80. Moreover, if one applies the McMurdo delta R of Hall et al. (2010) to McMurdo Sound carbonates, which seems more appropriate given their proximity to the calibration site, then the sound must have opened before 7.5 ka (Licht et al., 1996 date from sound) to 6.5 ka (oldest shells in beach deposits along McMurdo Sound – Hall et al., 2004). This leaves interpretation of the GC80 core site with two options – 1) the 1480 yr reservoir correction is too large, or 2) the minimum-limiting date for open marine conditions does not closely constrain the timing of grounding line retreat (the authors make a case against this from the lack of sub ice-shelf facies).

Fig. 3 – this figure is difficult to follow. In panel A, one cannot distinguish between grounded and floating ice. In addition, the time slices given do not match those in the caption. For example, (iii) says <5.2 ka on the figure but 5.8-5.2 ka in the caption. What is the blue shading along the coast in (iii)?

Panel B – If the authors actually applied the smaller reservoir correction to their McKay and Mawson cores, I think they would find their data agree better with the cosmogenic ages of ice thinning.

The text needs more information on the exposure age data beyond a simple reference to the Jones papers. What production rate was used? Scaling? Were external errors used or just internal ones? Were ages recalculated to be consistent between the two datasets? Looking up the original papers, it appears that Borchers and LSDn were used in the Mawson paper. The Borchers et al. rate is almost certainly too high, based on rates from both New Zealand and South America. LSDn generally gives younger ages than St or Lm. Using Borchers and LSDn then gives a young combination for the exposure ages

meaning that if one chose a different production rate or scaling scheme, ages would get older. The point here is that these exposure ages are about as young as they can be within existing production rate and scaling frameworks and thus any changes will likely bump them older, increasing rather than decreasing the gap between the proposed marine ages and the thinning history from the glaciers.

The Mawson data are very messy, and could give a wide range of thinning scenarios. McKay, on the other hand, is a beautiful dataset and shows the ice surface dropped like a rock at 7 ka, reaching the present glacier surface level at about that same time. This is quite a bit older than the 5.3 ka proposed age for the core near its mouth. A more realistic delta R would improve the data fit to the cosmogenic data, as well as to those two orange bands that represent modelled grounding line retreat.

For Mawson, I would mention that Mt. Murray is well inland of Bruce Pt and thus the ages are not surprisingly younger. Would it be more useful to just plot the Bruce Pt samples as those are ones with the closest relationship to the dates in the offshore core?

Fig. 3 Caption – please specify throughout whether you mean grounded ice or an ice shelf.
line 205 in caption – “results in good agreement between simulated and recorded ...” I’m not sure I would characterize it as good agreement – the modelled thinning history seems nowhere near as fast as the documented.

Line 212 – see Point 1 above

line 216 – see Point 1 above and comments for line 171

line 220 – what is the basis for this range of retreat dates? Give the basis here to remind the reader.

line 233-234 – elaborate – the McMurdo Sound shell and beach dates when corrected with the regular delta R align pretty well with the McKay Glacier cosmogenic record, so the authors might want to be specific about what did not align (e.g., the Coulman High core and other marine 14C records based on TOC).

261 – these dates (7.7-6.1 ka) don’t appear to come from your results. Based on Fig. 4C it looks as if they come from a recalculation (?) of the CH-2 core (mentioned in passing in the paper in captions up until this point) of McKay et al. and the Venturelli et al. Mercer Lake dates, which have not been mentioned at all in the text to this point. If retreat of the whole southern embayment (the main difference between 4B and 4C) is going to hinge on these two dates then these sites and data need to be adequately described and referenced.

261 – do you mean an embayment in the grounded ice sheet?

267 – unsure where these 7-5 ka ages appear from – based on what? Prior paragraph said 7.7-6.1. Overall, this section needs better explanation, referencing, and consistency of dates used.

269 – Actually, glacier surface lowering in the southern TAMS begins quite a bit earlier than the dates of the cores – e.g., Spector et al., 2017.

272 – the average reader will not know where these glaciers are so they should be in a map.

The 6 ka grounding line north of McMurdo Sound is not in accord with shell dates from the coast or from the floor of McMurdo Sound when a reasonable reservoir correction is used. And, it is not in agreement with the U/Th dates of coral from the ice shelf region.

There is no way to match the ages on fig 4C with the references that produced them and many have not been discussed at all in the text (e.g., Anderson, Stutz, Balco, Goehring etc...). These data need to be given somewhere (even if only in the SI).

What is the triangle dated to >1.7 ka under the WAIS? There is no triangle in the legend.

Not sure what is meant in the legend by “coastal lowering date”. Two of those appear to be relative sea-level curves, which give the timing of unloading and development of open marine conditions.

Dates of >5.8 and >5.4 for the Mawson and McKay core sites on fig. 4C contrast with how these data are presented in the text as the actual (rather than minimum) ages of grounding-line recession. I agree with the > symbol and suggest making the minimum nature of these ages for grounding-line recession more clear in the text. In addition, in line 223, these dates are listed as 5.9 and 5.3 ka. They should be consistent throughout.

line 283 – as mentioned above, I think recalibrating any of the dates from McMurdo Sound or Terra Nova Bay based on the 1480 yr delta R is not justifiable. Were the Venturelli et al dates also recalibrated? Table in SI lists the date for that site as 6.3 ka (as does the original publication), not >6.1 ka as given in fig. 4C.

line 540 – What thickness of core was required to arrive at those sample sizes? 1 cm? 5 cm? 10 cm? Give a sense as to the degree of time averaging captured by these dates.

line 602 – see Point 1

Supplemental Information

Please be careful throughout not to use the terms reservoir age and delta R interchangeably. They are very different things.

Please reference Extended Data 2 for the RP dates in the captions, so that the reader can find the original radiocarbon ages and metadata.

According to Ext Data table 2, the raw core top age for Split 2 of GC78 is 1440 yrs, which is less than the applied delta R of 1480 yrs (and the implied ocean reservoir of about 2080 yrs if mean ocean value is about 600 yrs). This alone demonstrates that the applied reservoir correction is too large by a lot. So, I do not understand why such a large delta R is applied here, when the authors' own data so explicitly contradict it. In contrast, use of the conventional 610 yr delta R gives an age of about 290 +/- 270 yrs at about the same depth as the 37 +/- 15 yr Pb sample, which is spot on the Pb data.

The core top radiocarbon age for GC80 is similarly too young to apply such a large delta R.

Text S2

You may want to reference Venturelli in this section.

Fig. 6 shows that while Split 2 likely contains the highest proportion of autochthonous carbon (line 165), it does not show that all of the sediment is autochthonous. A non-zero percent (looks like 10-12% in GC80) is likely reworked and another substantial percent, nearly half, could be either autochthonous or reworked, according to Table S2 and the text. To me, this reads that for core GC80, only roughly a third of the compounds can confidently be assigned an autochthonous origin likely of the correct age (although I suspect even that cannot account for resuspension and resettling of local material) and even taking everything except the PAHs to be autochthonous still gives ~10% refractory and likely old carbon contamination. Thus, any radiocarbon ages are maximum ages only. This point is stated in lines 225-226. But this concept vanishes in the main paper and the ramifications of this fact for calculating the reservoir age are not appreciated.

Fig S6 caption – are these data for core-top sediments? What is the depth?

Text S3

Line 236 – the terms R, delta R, and reservoir are used inconsistently. This is the first place R is used and may confuse a reader. It is unclear on line 236 what the R is that is referred to – not the corals (R= 1144) and not the R calculated at site 72 (which would be over 2000 yrs). I suggest just using delta R throughout the paper.

Lines 233 – see Point 1

Lines 240-241 – The timing is highly sensitive to the reservoir age and a correct reservoir age is necessary to make meaningful onshore-offshore comparisons, which is one of the stated goals of this paper. It does have the potential to change the narrative of retreat.

Table S3 –

It is unclear how the ref 14 coral reservoir effect was calculated in this table. It looks as if 603 yrs was simply subtracted from the published R value and the same error used. This is incorrect. This 1144 yr age offset was calculated with an early version of the Marine radiocarbon database and cannot be used with Marine20. The original calibration data have subsequently been recalculated by at least two studies and new delta Rs specific to Marine20 have been presented in both Gao et al., 2022 (609 +/- 137 yr) and Hall et al., 2023 (610 +/- 110 yr). The Marine20 compliant delta R also can be calculated from the Marine Chrono database.

4.3 ka age on fig. 4C is not in the table

Line 288-289 – units?

305-307 – I would agree with “provides a more reliable local marine reservoir correction” than some of the previous attempts using dates of bulk TOC. However, that does not mean it is reliable enough to produce a reservoir correction that is then applied widely. Closer to the true value does not mean correct.

312 – The production rate and scaling used need to be presented here.

Reviewer #2

(Remarks to the Author)

The work is very interesting and presents significant results, making it certainly worthy of publication following minor to moderate revision.

The issue of deglaciation after the Last Glacial Maximum (LGM) in the Ross Sea -marked by the retreat of the grounding line of the marine-based Ross Ice Shelf, which extended to the edge of the continental shelf- has been widely debated and remains a central topic of interest. This is because past changes in the grounding zone are critical for understanding the vulnerability and resilience of ice sheets to climate change and to variations in ocean temperature and current circulation. Several models have been proposed to reconstruct the retreat phases of the grounding line in the Ross Sea following the LGM. Among them are the “swinging gate” model by Conway et al. (1999), which suggests a general retreat across all troughs, and the “saloon door” model by Ackert (2008), which hypothesises that different troughs experienced distinct retreat phases. More recently, Halberstadt et al. (2016) proposed a marine-based model characterized by nine retreat stages of the Antarctic Ice Sheet, constrained by numerous radiocarbon dates (see also Prothro et al., 2020).

Radiocarbon dates obtained from marine organisms provide a wealth of information for dating the deglaciation phases that accompanied relative sea-level rise and the emergence of ice-free areas. These newly exposed coastal regions became available for recolonization by penguins and elephant seals following the LGM retreat of the glacial system along the Victoria Land coast (e.g., Baroni and Hall, 2004; Hall et al., 2004, 2006, 2023; Gao et al., 2022).

Recent studies suggest that the retreat phase was interrupted by one or more re-advances of the grounding line during the Holocene (e.g., Greenwood et al., 2018), and likely also during the Antarctic Cold Reversal (Baroni et al., 2022; Giorgetti et al., 2024).

These studies highlight the strongly regional behaviour of the glacial system involved in the progressive retreat and

deglaciation of coastal areas. However, the primary challenge facing all research on Antarctic deglaciation lies in accurately constraining the timing of these phases. In particular, there are difficulties in correlating ages derived from marine sediment cores with radiocarbon dates from marine-derived materials collected in deglaciated coastal areas—such as shells from Holocene raised beaches, and penguin remains from abandoned colonies and nesting sites.

Among the strengths of this work, I would like to highlight the following:

a) The authors' effort to correlate the timing of events recorded in marine sediments with evidence of post-LGM glacial retreat on land—documented through thinning of outlet glaciers along the Victoria Land coast and progressive deglaciation of coastal areas—is commendable and significantly contributes to the understanding of regional ice dynamics.

b) The application of ramped pyrolysis oxidation (RPO) radiocarbon (^{14}C) dating is particularly interesting. This method, which thermochemically separates different carbon pools in the sediment, addresses several limitations related to contamination by old carbon in marine sediments.

Additionally, the combined use of ^{210}Pb and RPO ^{14}C dating provides valuable results for estimating reservoir ages in sediment cores.

Among the weakness of this work, I indicate the following:

a) Authors state that the “reconstructed pattern of deglaciation is supported by regional-scale modelling in the SW Ross Sea, that shows glacial retreat is initially slow as the grounding line is pinned to shallow bathymetry to the north of Ross Island, but then retreat accelerates into deeper water”: actually, I did not find where results coming from modelling are presented in the paper to really support the reconstruction here proposed.

b) Reconstruction of deglaciation phases presented in figure 4 is well depicted but would need a check of the calibrated dates obtained from different DRs coming from different materials (see suggestion for improvement).

c) Authors exclude that “marine and terrestrial ice sheets deglaciated independently”, because this is not consistent with their “understanding of the relationship between grounding line retreat and upstream dynamic thinning of outlet glaciers”. Actually, grounding line retreat occurred at different velocity and at different time to the west and to the East of Ross Island, as clearly stated by ages from foraminifera collected in core CH-2 (Fig. 1) and dates coming from McMurdo Sound (both on land as indicated by Hall et al., 2004, 2006, 2023; and from ages coming from RPO dates here presented).

Suggestions for Improvement

To improve the manuscript, I suggest that the authors further clarify the following key issues, in addition to addressing the list of minor comments provided below.

The primary concern, in my opinion, relates to the correction of radiocarbon ages from marine sediments and the calculation of ΔR values used for calibration of radiocarbon dates from different organic materials. Specifically, I question whether the reservoir age derived from marine sediments can be reliably used to calculate ΔR values for marine-living organisms, which yield significantly different values (e.g., Hall et al., 2010, ref. 419; Gao et al., 2022, ref. 40).

Furthermore, the authors should discuss whether the reservoir age they obtained has local or regional applicability. In other words, can this reservoir age be applied consistently across areas ranging from Coulman Island to Ross Island, and to the entire dataset of radiocarbon dates considered in the study? Please elaborate on this point and, if necessary, consider applying different ΔR values for different types of organic materials (specifically dates from TOC sampled in sediment cores and dates from marine-living organisms collected on coastal deglaciated areas).

About results from modelling of glacier retreat I suggest revising this part and furnish more information on this point or hold down the support furnished by the model to the proposed reconstruction of glacial retreat.

As concerns grounding line retreat to the East and to the West of Ross Island, I suggest to revise the discussion taking into account the ages coming from all the cores, including dates from CH-2.

Specific issues

Rows 162-163: “Our RPO reservoir age is slightly older than that reported from the western Ross Sea based on carbonate dates from corals (ΔR of 609 years_{40,41}).”

Actually, ΔR of 609 ± 137 furnished by Gao et al 2022 (ref. 40) was obtained by:

a) conventional ^{14}C dates of known-age marine-life samples from the Ross Sea (penguins and seals of known age collected during the heroic age of Antarctic exploration) ($n = 13$); and

b) paired ^{14}C and U/Th ages of coral samples from Terra Nova Bay and McMurdo Ice Shelf of Ross Sea (Hall et al., 2010, ref. 41) ($n = 39$).

The total number of samples used for calculating DR to insert in Marine20 CALIB program is 52.

In my opinion, DR calculated by Gao et al (2022) should be used for calibrating samples collected on coastal ice-free areas, and to calibrate ages of marine-living organism.

Different is the case for RPO ages obtained from marine sediments cores, for which the DR obtained using RPO analysis is a new relevant contribute to correct and to calibrate the ages obtained from TOC.

Fig. 1: Please, insert geographic coordinates and toponyms of sites cited in the text (e.g. Siple Coast).

Fig. 2: Please, insert geographic coordinates in “A”.

Supplementary Information

The supporting information includes extended data for core sites GC71, 72, 78 and 80 (described in detail in Figure S1-4).

Please, insert coordinates and depth of coring site as well as other information on the cruise (for each site in the figure caption or, better, listed in supplementary table).

References

- Ackert, 2008. Swinging gate or Saloon doors: Do we need a new model of Ross Sea deglaciation? Fifteenth West Antarctic Ice Sheet Meeting, Sterling, Virginia.
- Anderson et al., 2014. Ross Sea paleo-ice sheet drainage and deglacial history during and since the LGM. *Quaternary Science Review*, 100, 31-54. Doi: 10.1016/j.quascirev.2013.08.020
- Baroni et al., 2022. Antarctic Ice Sheet re-advance during the Antarctic Cold Reversal identified in the Western Ross Sea. *GFDQ*, 45 (1), 3-18. <https://doi.org/10.4461/GFDQ.2022.45.1>
- Conway et al., 1999. Past and Future Grounding line Retreat of the West Antarctic Ice Sheet. *Science*, 286 (5438), 280-283. doi: 10.1126/science.286.5438.280
- Giorgetti et al., 2024. Post-LGM glaciomarine processes revealed by inner shelf sedimentary facies analysis (Terra Nova Bay, Western Ross Sea, Antarctica). *Quaternary International*, 697, 64-77. <https://doi.org/10.1016/j.quaint.2024.06.012>
- Greenwood et al., 2018. Holocene reconfiguration and readvance of the East Antarctic Ice Sheet. *Nature Communications*, 9:3176. Doi: 10.1038/s41467-018-05625-3
- Halberstadt et al., 2016. Past ice-sheet behaviour: Retreat scenarios and changing controls in the Ross Sea, Antarctica. *Cryosphere* 10, 1003-1020. Doi: 10.5194/tc-10-1003-2016
- Hall et al., 2006. Holocene elephant seal distribution implies warmer-than-present climate in the Ross Sea. *Proceedings of the National Academy of Sciences of the United States of America*, 103 (27), 10213-10217. doi:10.1073/pnas.0604002103
- Hall et al., 2023. Widespread southern elephant seal occupation of the Victoria Land Coast implies a warmer-than-present Ross Sea in the mid-to-late Holocene. *Quaternary Science Reviews*, 303, 107991. doi: 10.1016/j.quascirev.2023.107991
- Prothro et al., 2020. Timing and pathways of East Antarctic Ice Sheet retreat. *Quaternary Science Reviews*, 230 (106166), 1-20. Doi: 10.1016/j.quascirev.2020.106166

Reviewer #3

(Remarks to the Author)

Version 1:

Reviewer comments:

Reviewer #1

(Remarks to the Author)

I reviewed this paper previously and had a lot of comments on that initial draft. In my opinion, the authors have made a good-faith effort to thoroughly address those comments, and I have no further changes to suggest.

Reviewer #2

(Remarks to the Author)

The authors have addressed all the suggestions and clarified the points I previously indicated as unclear or problematic. In particular, I appreciate the use of a regional ΔR applied within the Calib program, with the exception of two cases (GC71 and GC72), where the authors applied an additional local contamination offset of 870 years to account for a significantly older core-top ^{14}C age obtained from paired ^{210}Pb data (site-specific correction).

Regarding the regional-scale modeling applied in this study, the authors have better clarified that they used model results previously published in combination with newly provided retreat ages from this paper; they have modified the text to better explain this point. Furthermore, they have added a figure to the supplementary information (Fig. S7) to illustrate the relationship between modeled and recorded retreat in the southwestern Ross Sea.

The section titled 'Inland ice drawdown coupled to rapid grounding line retreat' has been revised to provide a more comprehensive discussion of glacier retreat modeling in the context of the new retreat ages presented here, along with supporting thinning chronologies, to substantiate the proposed reconstruction.

The discussion has been extensively revised (lines 281-290), and core site CH-2 has been incorporated into Fig. 3 to enable a comprehensive comparison of retreat timing across all sites surrounding Ross Island.

Specific issues have been addressed, and corresponding modifications have been made to the text and figures accordingly.

Regarding the inserted citations, please note that citation number 55 (Baroni et al.) refers to volume 45 (not 43) of *Geografia Fisica e Dinamica Quaternaria*, and the publication year is 2022 (not 2023).

Response to reviewers

Below we provide full details on all changes made to our manuscript and point-by-point responses to all points raised by all reviewers (blue text).

REVIEWER COMMENTS

Reviewer #1 (Remarks to the Author):

The authors present here new marine radiocarbon data based on ramped pyrolysis (RP) from the western Ross Sea that bear on the timing of ice recession following the last glacial maximum. They integrate these data with others from the region, as well as cosmogenic nuclide ages of ice thinning of nearby outlet glaciers, to present a synthesis of the timing and pattern of Holocene ice retreat.

The authors are to be commended in their attempt to bring together a wide range of different data types in their synthesis. However, this paper suffers from issues that should be addressed prior to publication.

We thank Reviewer 1 for their thoughtful and constructive feedback. We appreciate the recognition of our effort to synthesise multiple datasets, including new radiocarbon results, previously published marine records, and cosmogenic nuclide data, to better constrain Holocene ice retreat in the western Ross Sea.

We also acknowledge the concerns raised and are particularly grateful for the time this reviewer has invested to engage deeply with our work. We have thoroughly revised the manuscript to address both general and specific issues identified in the following comments, and we believe these changes have made it a stronger paper. The point-by-point response below details our response to each recommendation.

In my opinion, the main issues can be summarized as:

- 1) The new delta R is not justifiable both in its calculation and application. It also leads to inconsistencies in the dataset that would not exist if they used a standard delta R for this region (which is well-known).
- 2) The paper needs more specific information on how calculations were done and on how ages presented in the text were determined. There are also numerous inconsistencies and/or typos in the text and figures that need resolving prior to publication.

We thank the reviewer for highlighting the need for greater clarity and consistency in our methodological descriptions and data presentation. We have made substantial revisions throughout the manuscript to address these issues, including a careful review and correction of inconsistencies between figure captions and figures, and between text and tables. All referenced data now align across text, tables, and figures, and we have also double-checked for formatting and citation errors.

1. Delta R

Marine dates in this study are based on ramped pyrolysis (RP), which aims to separate out different carbon fractions that have different origins, with the thought being that the more labile fractions reflect more recent marine carbon deposition and the more refractory components issuing from reworked ancient material. Ideally, one would be able to isolate only the recent fraction in order to obtain accurate ages for Ross Sea sediments. Dating has come a long way since the early studies in this region using simple TOC – which produced grossly old (and erroneous) ages for deglaciation. However, RP is not perfect and 100%, confident separation of the fractions cannot at present be achieved. This has been noted in prior studies (e.g., Venturelli et al., 2023) and appears to be the case in this present study. In the Supplemental Information (SI) the authors do a good job of explaining which fractions are thought to be autochthonous and which are reworked. There is also a substantial number of compounds that could be from both sources. Table S2 is especially helpful. However, from a combination of Table S2 and Fig. S6, one can see that even in the preferred Split 2, only roughly half of the compounds can be confidently attributed to autochthonous sources. Another ~40% could be either and about 7-10% are from refractory sources (ancient). Thus, the resulting radiocarbon dates from this split have a likely component of old material of at least 7-10% and maybe as much as 50%. This must affect the age, which means that these RP ages should be considered maximum ages only. Lines 225-226 of the SI refer to the resulting radiocarbon ages correctly as “maximum ages,” a point that seems lost when calculating and applying the delta R in the main paper.

If these RP ages are only maximum limiting ages, then they cannot be used in the manner proposed here to derive a marine delta R, because they are not the true radiocarbon age of the core top. At best, it is a maximum possible delta R for that place and time. As the proposed delta R (1480 yrs) calculated by this dubious manner is more than twice that of other measured values in the area (~610 yr; recalculated in Gao et al., 2022 and Hall et al., 2023) – in an area that already has one of the largest delta Rs in the world – it seems likely that the delta R proposed here is too large and should not be used.

The authors suggest that the commonly used delta R value of ~610 yr for McMurdo Sound (site of the original Hall et al. 2010 U/Th measurements from corals) may be younger than that for their site east of Ross Island due to ocean circulation

differences. However, similar delta R values also exist for Terra Nova Bay to the north (Hall et al., 2010), near the Siple Coast grounding line (Venturelli et al., 2023) nearly 1000 km away, and in the entire circum-Antarctic region based on marine shells collected prior to 1940 (when recalculated; Berkman and Forman, 1996), suggesting that this ~610 yr delta R value has robust regional significance. A quick look at the Marine Chrono database reveals that the average delta R for all Ross Sea samples is ~620 yrs. Thus, this delta R of 1480 yr calculated here is more than 3 sd from the mean and needs to be reevaluated critically. The authors do at one point mention that their delta R might be only applicable to their local site, but they then nevertheless apply it widely in the region, including to their two core sites on the western coast, one of which (McKay) is at the north end of McMurdo Sound. Applying their large delta R to these two core sites (GC78 and 80) results in the core top ages dating well into the future. All of this suggests that the proposed delta R is simply too large, most likely because no matter how good the RP and how much of an improvement over prior methods (which it is), there is some component of old material still left, which is affecting the age, making the delta R calculation an overestimate.

The authors suggest that the exact ages (which differ by nearly 1000 yrs) don't matter, just the pattern. I disagree. The pattern has been emerging for some time, with early models of a "swinging gate" giving way over the last decade to "saloon doors" with the center of the embayment opening up first. The approximate timing of retreat – mid-Holocene event – also has been known for some time. It is at the point where we need precise chronology and reliable onshore-offshore correlations – which is why exact ages do matter.

We thank the reviewer for their careful analysis of our approach on our use of RPO dates and the calculation and application of local marine reservoir corrections (ΔR). Based on this feedback, we have revisited our approach to ΔR for these sites. Our revised manuscript now adopts an estimate of ΔR of 609 ± 137 years from Hall et al. (2010) and Gao et al. (2022), and includes a discussion of how we approach the much older core top ages for GC71 and GC72 (Text S3).

We also agree that RPO, while a significant improvement over bulk TOC dating, still cannot perfectly isolate purely autochthonous fractions, and that some residual contamination from older, reworked carbon may remain. We have updated the main text and SI to better reflect this limitation and its implications for our chronological interpretations.

The revised text emphasises that even though our preferred fraction (split 2) has been demonstrated to concentrate primary photosynthetic products based Py-GC-MS analysis, it still contains a mixture of carbon sources. This includes a fraction of material of reworked or ambiguous origin, as shown by the compound classification table (Table S2) and Py-GC-MS data (Fig. S6). Accordingly, we treat our calibrated split 2 RPO ^{14}C dates as *best maximum ages* and have made this treatment more explicit throughout the manuscript. We also clarify that our ΔR estimate is derived from a comparison between the maximum age from ^{14}C and ^{210}Pb date from GC72 at 10 cm depth, which is not applied across all cores (e.g. lines 145-146 in main text; '*...and a reverse-calibrated pre-bomb ^{210}Pb - ^{14}C pair in GC72 at 10 cm depth^{40,41}*' and line 256 in SI; '*We reverse-calibrated a paired pre-bomb ^{210}Pb and ^{14}C date from GC72 at 10 cm depth*').

In response to the specific concern about the derived ΔR of 1480 year being too large for broad application, we have revised the manuscript and SI to clearly state that this value includes a local contamination offset, of 870 years, which is the difference between our ΔR value and the regional ΔR of 609 ± 137 years. It is unique to cores GC71 and GC72, rather than a revised regional ΔR . This offset was calculated only to correct those two cores, based on a pre-bomb ^{210}Pb - ^{14}C pair, of which no pair exists for GC78 and GC80. As described above, we now apply the regional ΔR of 609 ± 137 years (compiled by Gao et al., 2022) throughout our analysis, including all of our cores. GC78 and GC80 show smaller offsets between ^{210}Pb and ^{14}C dates, and we avoid applying any additional correction to them. We exclude the surface ^{14}C dates in these cases (post-bomb) and instead rely on the ^{210}Pb for upper age control. This revised approach is explained in the main text (lines 128-153 and the SI (Text S1 and S3).

In response to the final comment above – we agree that exact timing is important and the text in question has been removed.

2. Need for specific information and correction of inconsistencies.

This paper needs more detailed methodological information. Readers are left at critical points wondering exactly how certain ages were calculated, whether the number referred to is actually a delta R (as opposed to a reservoir effect), what production rates and scaling were used for cosmogenic ages, were internal or external errors used in comparisons, etc... There are also several instances of figures not matching captions and text not matching tables. Line examples for all of these are below.

Thank you for highlighting this here and below. In response, we have made substantial revisions throughout the manuscript to move key methodological details from the SI into the main text to improve transparency and accessibility for readers. We have clarified our use of the ΔR value and reservoir effect throughout (e.g. lines 129-130: '*...calibrated using the Marine20 curve and a regional marine reservoir correction (ΔR) of 609 ± 137 years^{33,39}*') and included further detail on

the cosmogenic ages (e.g. Methods section 'Glacier thinning from cosmogenic exposure dating'). See detailed responses below.

We have also clarified our terminology regarding marine radiocarbon calibration, specifically the applied ΔR value, and now use this terminology consistently throughout the manuscript (e.g. line 129-130, 142, 145, 150 and in the methods section 'Age model development'). In addition, we have added specific detail on our approach to cosmogenic nuclide exposure dating, including scaling scheme, production rates and the use of internal vs. external uncertainties in age comparisons. See response below for full details.

Where inconsistencies were identified (e.g. figure captions, table references and text), we have corrected these to ensure alignment across the manuscript and SI.

Line comments:

35-36 – I am not sure that the results as interpreted here reconcile long-standing differences. In fact, by introducing such a large reservoir effect, the marine ages are now younger, in my opinion, than the cosmogenic ages at McKay Glacier or the coastal marine deposits of McMurdo Sound.

Following our adoption of the Gao et al. ΔR value, our results do agree well, which is illustrated in the updated Fig. 3. The retreat age of GC80 (located at the head of the overdeepened trough of Mackay Glacier), aligns nicely with the rapid glacier lowering of Mackay Glacier.

69 – Is Ross Island really the southwestern Ross Sea? There is a lot more Ross Sea south of Ross Island.

"Southwestern Ross Sea" is a common geographic descriptor for the sector of the Ross Sea around Ross Island, bounded to the south by the front of the Ross Ice Shelf. However, we take the point that, when discussing the retreat timing of ice sheets and ice shelves, the entire Ross Sea (including areas presently covered by the ice shelf) should be acknowledged. We have updated the text in line 110 to reflect this but elect to retain "southwestern Ross Sea" elsewhere given its familiarity to our expected readers.

98 – this reference is out of date, with more recent papers and the Marine Chrono database suggesting 1100 yrs (e.g., Hall et al., 2010).

Updated to reference the more recent papers of Hall et al. (2010) and Gao et al. (2022) in lines 93: '*An additional challenge of dating marine sediments in the Antarctic is the large (~1100 years^{23,33}) and variable³⁴ reservoir age*'

Fig. 1 – needs lat./long.

Good spot – lat/long now added.

Fig. 2 – larger text would make this more readable

Font increased from 5 to 7 pt.

159 – I assume from the extended data table that the authors compared the samples at 10 cm depth and not at the core top. There needs to be specific information about which two dates were compared, if not here then definitely in the SI. This gets back to Point 2 above.

Yes we compared the ^{210}Pb and ^{14}C ages from 10 cm depth and have made this clearer throughout the manuscript (e.g. lines 145-146: '*... ΔR and a reverse-calibrated pre-bomb ^{210}Pb - ^{14}C pair in GC72 at 10 cm depth^{40,41}*').

160 – how did you reverse calibrate? Neither of the cited references give you how. Perhaps cite Reimer and Reimer 2017 if you followed their method.

We reverse calibrated the ^{14}C age by inputting the ^{14}C age and ^{210}Pb age (known age) at the same depth into the ΔR calculator (<http://calib.org/deltar>). This is the correct citation, but we now also cite Reimer and Reimer (2017), which

details this method (main text lines 145-146: '*...ΔR and a reverse-calibrated pre-bomb ²¹⁰Pb-¹⁴C pair in GC72 at 10 cm depth^{40,41}*' and SI Text S3).

161-163 As mentioned, this delta R should be treated with a large dose of skepticism. It is not "slightly older" than that reported from the corals of 609 yr. It's 900 yrs larger, which in the world of deltas R is a massive difference.

The reviewer's skepticism is valid, and we believe that revising the manuscript to address it has made the paper stronger. As detailed in our response to Point (1), we have overhauled this paragraph to reflect the recalibration and re-evaluation of our age models (e.g. lines 128-153). As such, we have removed this and other text associated with the application of the reservoir age we used in the original submission.

165 – how can you be sure of minimal contamination when the fraction that is confidently autochthonous is relatively small and there is a non-zero refractory component, which is almost certainly old?

As above, we have revised this paragraph to acknowledge that RPO reduces but does not entirely remove the influence of reworked carbon. As highlighted in the reviewer comment, we cannot be sure of minimal contamination, only that there is less old carbon contamination than bulk TOC ¹⁴C-derived data.

168 – this argument would be more convincing if similar reservoir effects to McMurdo Sound were not seen elsewhere (Point 1 above) = This implies to me that the reservoir age from site GC72 is either too old because of contamination by old material or that it is very site and time specific and should not be applied elsewhere.

Though we view it as an interesting possibility for future investigation, we have removed speculative text discussing the ways that differences in ocean water masses, including waters emerging from the ice shelf cavity, could be influencing the reservoir age in the SW Ross Sea region. As per our response to Point (1), the large reservoir age generated at site GC72 reflects local contamination, calculated as 870 years, which is the difference between a locally calculated ΔR value (based on the paired ¹⁴C and ²¹⁰Pb date from GC72) and the regional ΔR of 609 ± 137 years (e.g. lines 144-148: '*To correct for this, we apply an additional local contamination offset derived from the difference between the regional Ross Sea ΔR and a reverse-calibrated pre-bomb ²¹⁰Pb-¹⁴C pair in GC72 at 10 cm depth^{40,41}. The resulting local contamination of 870 ± 120 years was also applied to GC71 for which no equivalent pair exists (Supplementary Text 1, 3).*'

171 – Because I am not confident in the applied reservoir correction, I am not confident in the ages in this paragraph for ice retreat.

We have recalibrated and re-evaluated the ΔR value as suggested, resulting in changes to the ages of retreat at each core site. This paragraph is extensively revised.

I think it is also important to stress that your dates are for the onset of open marine conditions, which is not necessarily that of grounding-line retreat. It's a minimum age for grounding-line retreat. This point gets lost at various stages of the paper. Perhaps the use of "before" rather than "by" (e.g., line 181, 182) would help make this point. Hall et al. (2010) showed that coral was already growing in southern McMurdo Sound by ~ 6 ka from U-series dating. Thus, McMurdo Sound had opened up before that time, which is older than the dates you propose for sites 78 and 80. Moreover, if one applies the McMurdo delta R of Hall et al. (2010) to McMurdo Sound carbonates, which seems more appropriate given their proximity to the calibration site, then the sound must have opened before 7.5 ka (Licht et al., 1996 date from sound) to 6.5 ka (oldest shells in beach deposits along McMurdo Sound – Hall et al., 2004). This leaves interpretation of the GC80 core site with two options – 1) the 1480 yr reservoir correction is too large, or 2) the minimum-limiting date for open marine conditions does not closely constrain the timing of grounding line retreat (the authors make a case against this from the lack of sub ice-shelf facies).

Thank you for this comment. We agree that the dates reflect the onset of open water conditions, rather than grounding line retreat itself. We have updated the text to add clarity: '*open marine conditions were established by...*' (e.g. lines 159, 161-162, 164-165).

Following the recalibrated and updated age models, the minimum calibrated age for grounding line retreat at site GC78 is 6.9 (6.7-7.1 95% CI) cal yr BP and at site GC80 is 6.5 (6.3-7.3 95% CI) cal yr BP. The opening of McMurdo Sound before 7.5 (reworked shell; Licht et al., 1996) to ~6ka (based on the U-series dating from corals; Hall et al., 2010) fits nicely with our revised ages of retreat.

Fig. 3 – this figure is difficult to follow. In panel A, one cannot distinguish between grounded and floating ice. In addition, the time slices given do not match those in the caption. For example, (iii) says <5.2 ka on the figure but 5.8-5.2 ka in the caption. What is the blue shading along the coast in (iii)?

Figure 3 is extensively updated. We acknowledge it is difficult to distinguish between grounded and floating ice – we have added hatched shading to denote floating ice/ice shelf, which is over core site CH-2 and GC72 in both A(i) and (ii). The blue shading along the coast was used to denote ice free conditions (and therefore open ocean) along the coast following unloading of the ice sheet. We've changed this colour to light brown, representing submarine shoreface in areas where bathymetric information is not available. This ice/ocean interface has been difficult to represent in our schematics, but we hope these changes have made it clearer.

As the timing of open marine conditions at each core site has changed, the particular time slices we illustrate in A have also changed. We now show A(i) >7.2 cal kyr BP, when ice had not retreated beyond any of the core sites (older than the maximum 95% CI of all cores); A(ii) ~6.1 cal kyr BP when open marine conditions were established at core sites GC72, 78 and 80 (younger than 95% CI of these cores), but open marine conditions were not established at GC71 (so, older than the 95% CI of this core site); A(iii) <5.4 cal kyr BP when open marine conditions were present at all cores sites (younger than all 95% CI for all cores). We have made sure there is consistency between the time slices and the figure caption and made more explicit that we are referring to the 95% CI intervals, rather than the 'best' ages, which is more obvious in panel B(i) now.

Panel B – If the authors actually applied the smaller reservoir correction to their McKay and Mawson cores, I think they would find their data agree better with the cosmogenic ages of ice thinning.

The reviewer is absolutely right. After recalibrating the ^{14}C ages with the ΔR value of 609 ± 137 years, our marine retreat ages align nicely with the cosmogenic ages of ice thinning, particularly those of Mackay Glacier.

The text needs more information on the exposure age data beyond a simple reference to the Jones papers. What production rate was used? Scaling? Were external errors used or just internal ones? Were ages recalculated to be consistent between the two datasets? Looking up the original papers, it appears that Borchers and LSDn were used in the Mawson paper. The Borchers et al. rate is almost certainly too high, based on rates from both New Zealand and South America. LSDn generally gives younger ages than St or Lm. Using Borchers and LSDn then gives a young combination for the exposure ages meaning that if one chose a different production rate or scaling scheme, ages would get older. The point here is that these exposure ages are about as young as they can be within existing production rate and scaling frameworks and thus any changes will likely bump them older, increasing rather than decreasing the gap between the proposed marine ages and the thinning history from the glaciers.

The exposure ages from Mackay and Mawson glaciers were both calculated using the Borchers et al. (2016) production rate and LSDn scaling scheme, as reported in the Jones et al. (2020) paper, and ages with external errors (including production rate uncertainty) are used for comparisons in this study. We now explicitly state this information in the Methods 'Glacier thinning from cosmogenic exposure dating'.

The LSDn scaling scheme is the most up-to-date and accurate scaling scheme, and there is no reason to think that the older St or Lm schemes are equally as appropriate to use. But we acknowledge that the exact production rate for Antarctica is unknown, and using Southern Hemisphere calibration sites instead of a global calibration dataset scaled to Antarctica would make the exposure ages older. This was demonstrated in Jones et al. (2015) for Mackay Glacier. However, Antarctica has particular conditions (e.g. Stone, 2000), and there is no evidence that New Zealand or South America production rates are more suitable for Antarctica than the scaled global calibration dataset. There are also local influences on the production rate and resulting exposure age (e.g. erosion, glacial isostatic adjustment, snow cover) that, if corrected for, could feasibly make the exposure ages older; we discussed that this is unlikely in Text S4 (now in main text lines 238-256). For ^{10}Be in Antarctica, there is a higher chance that the ages are partially impacted by cosmogenic inheritance. Accounting for any inheritance would make the exposure ages younger, but we also consider this to be minimal. In summary, there are various factors that could potentially make the exposure ages younger or older, yet we consider these factors to be minimal. Nevertheless, we agree that this should be more fully acknowledged and have expanded the text in the main text (lines 238-256).

The Mawson data are very messy, and could give a wide range of thinning scenarios. McKay, on the other hand, is a beautiful dataset and shows the ice surface dropped like a rock at 7 ka, reaching the present glacier surface level at about that same time. This is quite a bit older than the 5.3 ka proposed age for the core near its mouth. A more realistic delta R would improve the data fit to the cosmogenic data, as well as to those two orange bands that represent modelled grounding line retreat.

Following recalibration and re-evaluation of the ΔR , the updated timing of retreat at GC80, near Mackay Glacier mouth, aligns very nicely with the cosmogenic data and orange bands that represent the modelled grounding line retreat.

For Mawson, I would mention that Mt. Murray is well inland of Bruce Pt and thus the ages are not surprisingly younger. Would it be more useful to just plot the Bruce Pt samples as those are ones with the closest relationship to the dates in the offshore core?

We have made the Mawson data grey with smaller symbols, and similarly changed the modelled ice surface elevation for Mawson Glacier to grey and in the background. By doing so, we still acknowledge Mawson Glacier and show that it is broadly consistent with Mackay Glacier data, but makes Mackay Glacier data the focus.

Fig. 3 Caption – please specify throughout whether you mean grounded ice or an ice shelf.

We have made this clearer throughout the caption for Fig. 3. Thanks.

line 205 in caption – “results in good agreement between simulated and recorded ...” I’m not sure I would characterize it as good agreement – the modelled thinning history seems nowhere near as fast as the documented.

We assume that this comment is referring to the rates of the modelled thinning vs. the rate of thinning implied by the cosmogenic data. This fit was made in the original paper by Jones et al. (2020) where the two experiments were very different, so it is understandably harder to see without that context. Compared to the rest of the deglaciation, there is a significantly faster rate of thinning in both the cosmogenic records and the model at the same time; the absolute rate of thinning may not be correct, likely due to uncertainty in the true climate forcing and model parameters.

Line 212 – see Point 1 above

Thanks for flagging – have updated to ‘*Combined with marine dates in the southwestern Ross Sea, recalibrated with the updated Marine20 curve⁴² and the Ross Sea regional $\Delta R^{33,39}$...*’ (lines 172-174)

line 216 – see Point 1 above and comments for line 171

As per our response to Point (1) and comment for line 171, we have re-calibrated the Coulman High grounded ice retreat date using the Ross Sea regional ΔR value of 609 ± 137 years, resulting in the minimum age of ungrounding at this site to be 8.6 (9.2-8.1, 95% confidence) cal kyr BP (lines 175-176 ‘*...minimum age of ungrounding at Coulman High¹⁰ becomes 8.6 (9.2-8.1, 95% confidence)...*’).

line 220 – what is the basis for this range of retreat dates? Give the basis here to remind the reader.

Have updated this sentence to ‘*Based on calibrated ¹⁴C ages following the transition from (sub)glacial to open marine facies in cores GC71 and 72, we constrain southwest retreat towards Ross Island to between 6.4 and 5.4 cal kyr BP.*’ (lines 181-183).

line 233-234 – elaborate – the McMurdo Sound shell and beach dates when corrected with the regular delta R align pretty well with the McKay Glacier cosmogenic record, so the authors might want to be specific about what did not align (e.g., the Coulman High core and other marine 14C records based on TOC).

We’ve added a sentence to be more specific about which marine dates have not previously aligned with terrestrial chronologies; ‘*This discrepancy is largest with the Coulman High core (8.6 cal kyr BP) and sites dated using the acid-insoluble organic fraction of bulk sediment*’ (lines 191-193).

261 – these dates (7.7-6.1 ka) don’t appear to come from your results. Based on Fig. 4C it looks as if they come from a recalculation (?) of the CH-2 core (mentioned in passing in the paper in captions up until this point) of McKay et al. and the Venturelli et al. Mercer Lake dates, which have not been mentioned at all in the text to this point. If retreat of the whole southern embayment (the main difference between 4B and 4C) is going to hinge on these two dates then these sites and data need to be adequately described and referenced.

Thank you for highlighting this area of confusion. We have carefully assessed the dates we refer to in this paragraph. We refer to the *minimum* statistically likely ages of retreat given by the 95% CI of each date/site in Fig. 4, and therefore use these minimum ages in this text. The dates (7.7-6.1) indeed come from the recalibration of CH-2 (Mckay et al., 2008) and the Mercer Lake dates (Venturelli et al., 2023), which are now updated to 8.1-6.2 cal kyr BP. We have added text about what sites we refer to, to make it clear for the reader – lines 284-286 ‘*Our results suggest that the embayment in grounded ice, bounded by Ross Island to the west and the Siple coast, formed rapidly, at the latest, between ~8.1 (site CH-2¹⁰) and 6.2 (Mercer Subglacial Lake²⁰) ka BP.*’

261 – do you mean an embayment in the grounded ice sheet?

Thank you for this clarifying question, we have updated the text (line 284).

267 – unsure where these 7-5 ka ages appear from – based on what? Prior paragraph said 7.7-6.1. Overall, this section needs better explanation, referencing, and consistency of dates used.

We have overhauled this paragraph as per the comment for line 261 (lines 280-290).

269 – Actually, glacier surface lowering in the southern TAMS begins quite a bit earlier than the dates of the cores – e.g., Spector et al., 2017.

Yes, surface lowering of southern TAMS glaciers (Reedy, Scott, Beardmore) occurs earlier than the dates of retreat recorded in the cores. This is consistent with our reconstruction of retreat in the Ross Embayment. We now clarify in the ‘Topography controls ice sheet retreat’ section (lines 302-315) that retreat would have occurred earlier in the central Ross Sea with corresponding surface lowering of glaciers in the southern TAMS, with delayed retreat in the SW Ross Sea due to the topographic control.

272 – the average reader will not know where these glaciers are so they should be in a map.

Thank you for highlighting. We acknowledge that the location of these glaciers may not be known by the average reader. We have added a table in the supplementary information (Table S4) that includes the latitude/longitude, age and reference for glaciers shown in Figure 4.

The 6 ka grounding line north of McMurdo Sound is not in accord with shell dates from the coast or from the floor of McMurdo Sound when a reasonable reservoir correction is used. And, it is not in agreement with the U/Th dates of coral from the ice shelf region.

We have updated the position of the 7 and 6 ka grounding lines in line with the recalibrated ages. We’ve added the recalibrated date from the reworked shell (Licht et al., 1996) – thanks for spotting this missing date.

There is no way to match the ages on fig 4C with the references that produced them and many have not been discussed at all in the text (e.g., Anderson, Stutz, Balco, Goehring etc...). These data need to be given somewhere (even if only in the SI).

As per the comment above for lines 272, we have added Table S4 to the supplementary information with the latitude/longitude, age and reference for each glacier shown in Figure 4. We refer to the TAM glaciers collectively in lines 305-306: ‘...with the exact timing varying between glaciers along the Transantarctic Mountains (Supplementary Table 4)...’.

What is the triangle dated to >1.7 ka under the WAIS? There is no triangle in the legend.

The triangle represented the grounding line retreat over the Bindschadler Ice Stream, Siple Coast from Neuhaus et al. (2021). However, we’ve now changed that symbol to a square, as this date reflects marine retreat. The date has also been updated to reflect the minimum statistically possible age of grounding line recession (see Table S3), in line with all of the marine retreat dates in this figure.

Not sure what is meant in the legend by “coastal lowering date”. Two of those appear to be relative sea-level curves, which give the timing of unloading and development of open marine conditions.

Thanks for spotting – we have updated the black circles to ‘unloading of grounded ice’ and the black inverted triangles to ‘glacier surface ice lowering’.

Dates of >5.8 and >5.4 for the Mawson and McKay core sites on fig. 4C contrast with how these data are presented in the text as the actual (rather than minimum) ages of grounding-line recession. I agree with the > symbol and suggest making the minimum nature of these ages for grounding-line recession more clear in the text. In addition, in line 223, these dates are listed as 5.9 and 5.3 ka. They should be consistent throughout.

We have added this sentence: ‘*Note the marine retreat ages are the minimum 95% confidence interval value, and therefore the youngest statistically likely timing of retreat at these sites (Supplementary Table 3)*’ to Figure 4 caption to make this point clearer. We hope that this distinction is now clearer.

line 283 – as mentioned above, I think recalibrating any of the dates from McMurdo Sound or Terra Nova Bay based on the 1480 yr delta R is not justifiable. Were the Venturelli et al dates also recalibrated? Table in SI lists the date for that site as 6.3 ka (as does the original publication), not >6.1 ka as given in fig. 4C.

Table S3 has been updated to reflect the recalibration of our own dates as described above.

We have not updated the dates from Venturelli et al. 2023 as the authors derived a local reservoir age from extant amphipods collected from the marine cavity at their site during borehole operations, which result in an average ^{14}C age of 1101 ± 38 years (Kingslake et al., 2018). They applied a standard deviation of ± 120 as recommended by Hall et al. (2010) to account for variation in the local reservoir in sub ice shelf and grounding line proximal environmental due to different water masses and DIC pools that are advected under the Ross Ice Shelf. The resulting local reservoir age used is 1101 ± 120 years, and using the Marine20 calibration curve, results in a ΔR value of $\sim 500 \pm 120$ years. This value is comparable to the Ross Sea average ΔR value.

Figure 4C shows the *minimum* statistically likely ages of retreat (based on 95% CI), which in the case of Venturelli is 6.2 ka, which we have updated in the figure to correct our mistake.

line 540 – What thickness of core was required to arrive at those sample sizes? 1 cm? 5 cm? 10 cm? Give a sense as to the degree of time averaging captured by these dates.

Sample thickness was 1 cm – have added detail to text (line 393).

line 602 – see Point 1

Supplemental Information

Please be careful throughout not to use the terms reservoir age and delta R interchangeably. They are very different things.

Thanks for pointing this out. We have made a conscious effort to make the distinction clear and refer to ΔR throughout the text, which prevents confusion with the reader also.

Please reference Extended Data 2 for the RP dates in the captions, so that the reader can find the original radiocarbon ages and metadata.

Done, thanks.

According to Ext Data table 2, the raw core top age for Split 2 of GC78 is 1440 yrs, which is less than the applied delta R of 1480 yrs (and the implied ocean reservoir of about 2080 yrs if mean ocean value is about 600 yrs). This alone demonstrates that the applied reservoir correction is too large by a lot. So, I do not understand why such a large delta R is applied here, when the authors’ own data so explicitly contradict it. In contrast, use of the conventional 610 yr delta R gives an age of about 290 ± 270 yrs at about the same depth as the 37 ± 15 yr Pb sample, which is spot on the Pb data.

The core top radiocarbon age for GC80 is similarly too young to apply such a large delta R.

Thank you for this helpful observation, which we agree with. As noted in our response to point (1), we now apply the regional Ross Sea ΔR rather than the previously larger correction. This approach is consistent with the ^{210}Pb constraints and avoids overestimation of the reservoir effect.

Text S2

You may want to reference Venturelli in this section.

We have chosen not to cite Venturelli et al. (2023) in this section as the work is based on RPO, not Py-GC-MS.

Fig. 6 shows that while Split 2 likely contains the highest proportion of autochthonous carbon (line 165), it does not show that all of the sediment is autochthonous. A non-zero percent (looks like 10-12% in GC80) is likely reworked and another substantial percent, nearly half, could be either autochthonous or reworked, according to Table S2 and the text. To me, this reads that for core GC80, only roughly a third of the compounds can confidently be assigned an autochthonous origin likely of the correct age (although I suspect even that cannot account for resuspension and resettling of local material) and even taking everything except the PAHs to be autochthonous still gives ~10% refractory and likely old carbon contamination. Thus, any radiocarbon ages are maximum ages only. This point is stated in lines 225-226. But this concept vanishes in the main paper and the ramifications of this fact for calculating the reservoir age are not appreciated.

We thank the reviewer for this careful observation and agree with their assessment. We now explicitly state throughout the main text that split 2 radiocarbon ages are *maximum ages* due to the presence of reworked and refractory carbon (e.g. lines 140-141: '*...we selected split 2 ages as providing the most accurate maximum age constraint for cores GC71, 72, 78 and 80*'). We also acknowledge that deconvolution modelling could help quantify autochthonous material in the split and assess age contribution from various carbon pools, and we look to explore those connections in future work.

Fig S6 caption – are these data for core-top sediments? What is the depth?

No, they're samples from downcore – the depths are in the figure and also now added to the figure caption.

Text S3

Line 236 – the terms R, delta R, and reservoir are used inconsistently. This is the first place R is used and may confuse a reader. It is unclear on line 236 what the R is that is referred to – not the corals (R= 1144) and not the R calculated at site 72 (which would be over 2000 yrs). I suggest just using delta R throughout the paper.

We have overhauled this text ('Reservoir correction and age model development') to reflect the changes we've made regarding the ΔR used to calibrate the ^{14}C dates and generate chronology. In doing so, we now only refer to ΔR as suggested throughout the text.

Lines 233 – see Point 1

Text S3 has been extensively revised. We refer to reservoir ages in the context of ΔR throughout the text now, not R in this case, as used in the original submission.

Lines 240-241 – The timing is highly sensitive to the reservoir age and a correct reservoir age is necessary to make meaningful onshore-offshore comparisons, which is one of the stated goals of this paper. It does have the potential to change the narrative of retreat.

We agree that the timing of retreat is highly sensitive to the choice of reservoir age. We have removed this sentence and clarified our use of the regional Ross Sea ΔR . This makes our age models more robust and allows a more meaningful comparison between onshore and offshore records.

Table S3 –

It is unclear how the ref 14 coral reservoir effect was calculated in this table. It looks as if 603 yrs was simply subtracted from the published R value and the same error used. This is incorrect. This 1144 yr age offset was calculated with an early version of the Marine radiocarbon database and cannot be used with Marine20. The original calibration data have subsequently been recalculated by at least two studies and new delta Rs specific to Marine20 have been presented in both

Gao et al., 2022 (609 +/- 137 yr) and Hall et al., 2023 (610 +/- 110 yr). The Marine20 compliant delta R also can be calculated from the Marine Chrono database.

All data has been recalibrated using the ΔR of 609 ± 137 years and the resulting calibrated ages are updated in Table S3.

4.3 ka age on fig. 4C is not in the table

Thanks for pointing this out – an oversight on our part. We have added this date into Table S3. This date is from the Whillans Subglacial Lake, Siple Coast from Neuhaus et al. (2021). It is modelled grounding line retreat derived from a comparison of the measured pore water ionic concentrations from a sediment core compared to modelled values using their ionic diffusion model. It is still a date reflecting marine retreat, and we have therefore changed the symbol to a square.

Line 288-289 – units?

Text S4 has been moved to the main text and the unit 'km' is added here (line 302).

305-307 – I would agree with “provides a more reliable local marine reservoir correction” than some of the previous attempts using dates of bulk TOC. However, that does not mean it is reliable enough to produce a reservoir correction that is then applied widely. Closer to the true value does not mean correct.

We agree. This text is updated and has moved to the main text on lines 233-236: *‘However, as discussed above, by combining ^{210}Pb data with RPO-derived ^{14}C ages, we reduce contamination by ancient detrital carbon and calibrate using the Marine20 curve and a Ross Sea average ΔR value of 609 ± 137 years^{33,39} to produce more accurate timings for the establishment of open marine conditions at the core sites.’*

312 – The production rate and scaling used need to be presented here.

We now describe how the cosmogenic ages were calculated, including the production rate and scaling used, in the Methods. We also discuss uncertainty in these ages in the main text, within the revised section ‘Inland ice drawdown coupled to rapid grounding line retreat’.

Text S4 has been moved to the main text under the section ‘Inland ice drawdown coupled to rapid grounding line retreat’.

Reviewer #2 (Remarks to the Author):

The work is very interesting and presents significant results, making it certainly worthy of publication following minor to moderate revision.

The issue of deglaciation after the Last Glacial Maximum (LGM) in the Ross Sea -marked by the retreat of the grounding line of the marine-based Ross Ice Shelf, which extended to the edge of the continental shelf- has been widely debated and remains a central topic of interest. This is because past changes in the grounding zone are critical for understanding the vulnerability and resilience of ice sheets to climate change and to variations in ocean temperature and current circulation. Several models have been proposed to reconstruct the retreat phases of the grounding line in the Ross Sea following the LGM. Among them are the "swinging gate" model by Conway et al. (1999), which suggests a general retreat across all troughs, and the "saloon door" model by Ackert (2008), which hypothesises that different troughs experienced distinct retreat phases. More recently, Halberstadt et al. (2016) proposed a marine-based model characterized by nine retreat stages of the Antarctic Ice Sheet, constrained by numerous radiocarbon dates (see also Prothro et al., 2020). Radiocarbon dates obtained from marine organisms provide a wealth of information for dating the deglaciation phases that accompanied relative sea-level rise and the emergence of ice-free areas. These newly exposed coastal regions became available for recolonization by penguins and elephant seals following the LGM retreat of the glacial system along the Victoria Land coast (e.g., Baroni and Hall, 2004; Hall et al., 2004, 2006, 2023; Gao et al., 2022).

Recent studies suggest that the retreat phase was interrupted by one or more re-advances of the grounding line during the Holocene (e.g., Greenwood et al., 2018), and likely also during the Antarctic Cold Reversal (Baroni et al., 2022; Giorgetti et al., 2024).

These studies highlight the strongly regional behaviour of the glacial system involved in the progressive retreat and deglaciation of coastal areas. However, the primary challenge facing all research on Antarctic deglaciation lies in accurately constraining the timing of these phases. In particular, there are difficulties in correlating ages derived from marine sediment cores with radiocarbon dates from marine-derived materials collected in deglaciated coastal areas -such as shells from Holocene raised beaches, and penguin remains from abandoned colonies and nesting sites.

Among the strengths of this work, I would like to highlight the following:

a) The authors' effort to correlate the timing of events recorded in marine sediments with evidence of post-LGM glacial retreat on land—documented through thinning of outlet glaciers along the Victoria Land coast and progressive deglaciation

of coastal areas—is commendable and significantly contributes to the understanding of regional ice dynamics.

b) The application of ramped pyrolysis oxidation (RPO) radiocarbon (^{14}C) dating is particularly interesting. This method, which thermochemically separates different carbon pools in the sediment, addresses several limitations related to contamination by old carbon in marine sediments.

Additionally, the combined use of ^{210}Pb and RPO ^{14}C dating provides valuable results for estimating reservoir ages in sediment cores.

We thank the reviewer for their thoughtful and positive evaluation of our manuscript. We also appreciate the recognition of the study's contribution to understanding post-LGM deglaciation dynamics in the Ross Sea region and the appreciation for the application of ^{210}Pb and RPO ^{14}C dating.

Among the weakness of this work, I indicate the following:

a) Authors state that the “reconstructed pattern of deglaciation is supported by regional-scale modelling in the SW Ross Sea, that shows glacial retreat is initially slow as the grounding line is pinned to shallow bathymetry to the north of Ross Island, but then retreat accelerates into deeper water”: actually, I did not find where results coming from modelling are presented in the paper to really support the reconstruction here proposed.

The regional-scale modelling was done in a previous study, and here we use the model results in combination with our new retreat ages to provide a physically-based explanation for the timing, pattern and mechanism of retreat. We have now made sure to cite the original study wherever we mention this modelling, have moved relevant discussion from Text S4 to the ‘Inland ice drawdown coupled to rapid grounding line retreat’ section in the main text, and have added a figure to the supplementary information (Fig. S7) to highlight the relationship between modelled and recorded retreat in the SW Ross Sea.

b) Reconstruction of deglaciation phases presented in figure 4 is well depicted but would need a check of the calibrated dates obtained from different DRs coming from different materials (see suggestion for improvement).

c) Authors exclude that “marine and terrestrial ice sheets deglaciated independently”, because this is not consistent with their “understanding of the relationship between grounding line retreat and upstream dynamic thinning of outlet glaciers”. Actually, grounding line retreat occurred at different velocity and at different time to the west and to the East of Ross Island, as clearly stated by ages from foraminifera collected in core CH-2 (Fig. 1) and dates coming from McMurdo Sound (both on land as indicated by Hall et al., 2004, 2006, 2023; and from ages coming from RPO dates here presented).

Suggestions for Improvement

To improve the manuscript, I suggest that the authors further clarify the following key issues, in addition to addressing the list of minor comments provided below.

The primary concern, in my opinion, relates to the correction of radiocarbon ages from marine sediments and the calculation of ΔR values used for calibration of radiocarbon dates from different organic materials. Specifically, I question whether the reservoir age derived from marine sediments can be reliably used to calculate ΔR values for marine-living organisms, which yield significantly different values (e.g., Hall et al., 2010, ref. 419; Gao et al., 2022, ref. 40).

Furthermore, the authors should discuss whether the reservoir age they obtained has local or regional applicability. In other words, can this reservoir age be applied consistently across areas ranging from Coulman Island to Ross Island, and to the entire dataset of radiocarbon dates considered in the study? Please elaborate on this point and, if necessary, consider applying different ΔR values for different types of organic materials (specifically dates from TOC sampled in sediment cores and dates from marine-living organisms collected on coastal deglaciated areas).

We agree that ΔR values derived from different archives can yield different results. In light of this, and as detailed in our response to Reviewer 1, we have revised our approach and now apply the most recent regionally constrained compiled Ross Sea ΔR value rather than relying solely on values derived from our ^{210}Pb -RPO ^{14}C data. We have clarified this in the methods and results sections and we no longer use our locally derived ΔR value for calibration of radiocarbon ages.

About results from modelling of glacier retreat I suggest revising this part and furnish more information on this point or hold down the support furnished by the model to the proposed reconstruction of glacial retreat.

The section ‘Inland ice drawdown coupled to rapid grounding line retreat’ has now been revised to more comprehensively discuss the modelling of glacier retreat in the context of our new retreat ages and supporting thinning chronologies to support our proposed reconstruction.

As concerns grounding line retreat to the East and to the West of Ross Island, I suggest to revise the discussion taking into account the ages coming from all the cores, including dates from CH-2.

The discussion has been thoroughly revised (lines 281-290) and CH-2 has also been added to Fig. 3, panel B to facilitate comparison of retreat timing across all sites around Ross Island.

Specific issues

Rows 162-163: "Our RPO reservoir age is slightly older than that reported from the western Ross Sea based on carbonate dates from corals (ΔR of 609 years^{40,41})."

Actually, ΔR of 609 ± 137 furnished by Gao et al 2022 (ref. 40) was obtained by:

- a) conventional ^{14}C dates of known-age marine-life samples from the Ross Sea (penguins and seals of known age collected during the heroic age of Antarctic exploration) ($n = 13$); and
- b) paired ^{14}C and U/Th ages of coral samples from Terra Nova Bay and McMurdo Ice Shelf of Ross Sea (Hall et al., 2010, ref. 41) ($n = 39$).

The total number of samples used for calculating DR to insert in Marine20 CALIB program is 52.

In my opinion, DR calculated by Gao et al (2022) should be used for calibrating samples collected on coastal ice-free areas, and to calibrate ages of marine-living organism.

Different is the case for RPO ages obtained from marine sediments cores, for which the DR obtained using RPO analysis is a new relevant contribute to correct and to calibrate the ages obtained from TOC.

We thank the reviewer for clarifying the methods used by Gao et al. (2022) to derive the ΔR of 609 ± 137 years from known-age samples, including marine-life specimens (penguins and seals) and paired ^{14}C -U/Th coral ages. We fully agree that this ΔR is an appropriate regional value for calibrating ages of modern marine organisms and materials from coastal ice-free areas. In our revised manuscript, we now state explicitly that this Gao et al. (2022) value is applied as the baseline ΔR for all of our marine sediment core calibrations, except in two cases (GC71 and GC72) where we applied an additional local contamination offset of 870 years to account for a much older core-top ^{14}C age relative to paired ^{210}Pb data. This offset is no longer intended to be used as a revised regional ΔR , but is instead a site-specific correction. For other cores (GC78, GC80), we did not observe a similarly large offset and so we rely on ^{210}Pb for surface age control, avoiding over-correction. We also clarify that our RPO-based ages are treated as *best maximum age* estimates due to potential residual reworked carbon, and we have expanded the text and SI to highlight how this uncertainty is incorporated into our interpretation (e.g. lines 128-153 and Text S3).

Fig. 1: Please, insert geographic coordinates and toponyms of sites cited in the text (e.g. Siple Coast).
Done, thanks.

Fig. 2: Please, insert geographic coordinates in "A".
Good spot - have added the coordinates.

Supplementary Information

The supporting information includes extended data for core sites GC71, 72, 78 and 80 (described in detail in Figure S1-4). Please, insert coordinates and depth of coring site as well as other information on the cruise (for each site in the figure caption or, better, listed in supplementary table).

We have added the cruise details (vessel, expedition and year) to the introduction of the Supplementary information (*R/V Araon, Expedition ANA05B-RS15, 2015*) to avoid duplication in the figure captions. The coordinates, depth of coring site and core length have been added to respective figure captions (Figures S1-S4).

References

- Ackert, 2008. Swinging gate or Saloon doors: Do we need a new model of Ross Sea deglaciation? Fifteenth West Antarctic Ice Sheet Meeting, Sterling, Virginia.
- Anderson et al., 2014. Ross Sea paleo-ice sheet drainage and deglacial history during and since the LGM. *Quaternary Science Review*, 100, 31-54. Doi: 10.1016/j.quascirev.2013.08.020
- Baroni et al., 2022. Antarctic Ice Sheet re-advance during the Antarctic Cold Reversal identified in the Western Ross Sea. *GFDQ*, 45 (1), 3-18. <https://doi.org/10.4461/GFDQ.2022.45.1>
- Conway et al., 1999. Past and Future Grounding line Retreat of the West Antarctic Ice Sheet. *Science*, 286 (5438), 280-283. doi: 10.1126/science.286.5438.280
- Giorgetti et al., 2024. Post-LGM glaciomarine processes revealed by inner shelf sedimentary facies analysis (Terra Nova Bay, Western Ross Sea, Antarctica). *Quaternary International*, 697, 64-77. <https://doi.org/10.1016/j.quaint.2024.06.012>
- Greenwood et al., 2018. Holocene reconfiguration and readvance of the East Antarctic Ice Sheet. *Nature Communications*, 9:3176. Doi: 10.1038/s41467-018-05625-3
- Halberstadt et al., 2016. Past ice-sheet behaviour: Retreat scenarios and changing controls in the Ross Sea, Antarctica.

Cryosphere 10, 1003-1020. Doi: 10.5194/tc-10-1003-2016

Hall et al., 2006. Holocene elephant seal distribution implies warmer-than-present climate in the Ross Sea. *Proceedings of the National Academy of Sciences of the United States of America*, 103 (27), 10213-10217. doi:10.1073/pnas.0604002103

Hall et al., 2023. Widespread southern elephant seal occupation of the Victoria Land Coast implies a warmer-than-present Ross Sea in the mid-to-late Holocene. *Quaternary Science Reviews*, 303, 107991. doi: 10.1016/j.quascirev.2023.107991

Prothro et al., 2020. Timing and pathways of East Antarctic Ice Sheet retreat. *Quaternary Science Reviews*, 230 (106166), 1-20. Doi: 10.1016/j.quascirev.2020.106166

Reviewer #3 (Remarks to the Author):

We thank Rev. 3 for reviewing our manuscript. We appreciate the role of Early Career Researchers in the peer review process and are grateful for the time and effort dedicated to evaluating our work.

Response to reviewers

Below we provide full details on all changes made to our manuscript and point-by-point responses to all points raised by all reviewers (blue text).

We are very grateful to our three reviewers for their highly constructive feedback, which has significantly strengthened our manuscript. Their thoughtful engagement with the data at a granular level exemplifies the best of peer review. We have made the requested corrections and clarifications, and we look forward to seeing the manuscript published in *Nature Communications*.

REVIEWERS' COMMENTS

Reviewer #1 (Remarks to the Author):

I reviewed this paper previously and had a lot of comments on that initial draft. In my opinion, the authors have made a good-faith effort to thoroughly address those comments, and I have no further changes to suggest.

We thank Rev. 1 for their careful consideration of our revisions. We are very pleased that our efforts to address the initial round of comments were found satisfactory.

Reviewer #2 (Remarks to the Author):

The authors have addressed all the suggestions and clarified the points I previously indicated as unclear or problematic.

In particular, I appreciate the use of a regional ΔR applied within the Calib program, with the exception of two cases (GC71 and GC72), where the authors applied an additional local contamination offset of 870 years to account for a significantly older core-top ^{14}C age obtained from paired ^{210}Pb data (site-specific correction). Regarding the regional-scale modeling applied in this study, the authors have better clarified that they used model results previously published in combination with newly provided retreat ages from this paper; they have modified the text to better explain this point. Furthermore, they have added a figure to the supplementary information (Fig. S7) to illustrate the relationship between modeled and recorded retreat in the southwestern Ross Sea.

The section titled 'Inland ice drawdown coupled to rapid grounding line retreat' has been revised to provide a more comprehensive discussion of glacier retreat modeling in the context of the new retreat ages presented here, along with supporting thinning chronologies, to substantiate the proposed reconstruction.

The discussion has been extensively revised (lines 281-290), and core site CH-2 has been incorporated into Fig. 3 to enable a comprehensive comparison of retreat timing across all sites surrounding Ross Island.

Specific issues have been addressed, and corresponding modifications have been made to the text and figures accordingly.

Regarding the inserted citations, please note that citation number 55 (Baroni et al.) refers to volume 45 (not 43) of *Geografia Fisica e Dinamica Quaternaria*, and the publication year is 2022 (not 2023).

We greatly appreciate Rev. 2's thoughtful engagement and detailed assessment of our revisions. We are glad that the clarifications and additional figures (e.g. Fig. S7) were found to strengthen the manuscript, and that the revised discussion and inclusion of core site CH-2 improved the comparison of retreat timing.

As noted, we have corrected the citation for Baroni et al. to volume 45 (not 43) and year 2022 (not 2023). We thank the reviewer for catching this error.